# Therapeutic targeting SPI1 in combination with erastin promotes ferroptosis in ccRCC
Wei Xue[1,3], Zhengqi Wu[1,3], Jixin Li[2], Peng Jin[1], Yuze Zhu[1], Zhiyuan Li[1], Zhijiao Wang[2], Zhenhua Li [1] ✉ & Xiang Fei [1] ✉

Clear cell renal cell carcinoma is the most common renal cell carcinoma subtype with a poor prognosis. The SPI1/Pu.1, which encodes a member of the E26-transformation-specific family of transcription factors, is highly expressed and associated with poor prognosis in renal cell carcinoma. Ferroptosis, a form of cell death distinct from apoptosis, pyroptosis, and necrosis, is characterized by iron accumulation and lipid peroxidation. Although the role of SPI1 in renal cell carcinoma is recognized, its relationship with ferroptosis remains unclear. In this study, we demonstrate that SPI1 is differentially overexpressed in renal cell carcinoma and associated with unfavorable prognosis. We also show that knockdown of SPI1 enhances erastin-induced ferroptosis. Furthermore, combining EZH2 inhibitors with erastin similarly promotes ferroptosis in renal cancer cells. Mechanistically, SPI1 transcriptionally suppresses ACSL4 expression through the EZH2/H3K27me3 pathway, leading to inhibition of intracellular lipid peroxidation. Thus, SPI1 knockdown synergizes with erastin to promote lipid peroxidation and ferroptosis, suggesting that targeting SPI1 may represent a promising therapeutic strategy for renal cell carcinoma.

Renal cell carcinoma (RCC) is one of the most common malignancies of the urinary system[1]. Clear cell renal cell carcinoma (ccRCC) is a common type of RCC that originates from the epithelial cells of the proximal renal tubules and accounts for more than 70% of renal malignancies[2]. Due to the subtle early symptoms of kidney cancer, many patients are diagnosed at an advanced stage, missing the optimal window for surgical intervention, with 20–30% already having distant metastases[3,4]. Unfortunately, kidney cancer is not sensitive to radiotherapy and chemotherapy[5], so immunotherapy, targeted therapy and finding effective biomarkers are what I am working on now.

SPI1/Pu.1 is a coding protein consisting of 272 amino acids, which encodes a member of the E26-transformation-specific family of transcription factors[6]. The SPI1 sequence contains different functional domains, including the ETs structural domain, which recognises the DNA sequence of the core GGAA[7]. It physically interacts with multiple regulatory factors to exert transcriptional effects, such as general transcription factors, haematopoietic-related transcription factors, erythrocyte factors and non-erythrocyte factors[8,9]. Transcriptional regulation is a complex and dynamic process, and the transcription factor SPI1 can bind chaperone molecules, co-activators and epigenetic molecules during transcriptional regulation[10,11].

Initially thought to be a transcriptional activator, SPI1 can bind to closed nucleosomes[12–14], increasing chromatin accessibility by recruiting epigenetic molecules like CBP/P300 or the SWI/SNF complex[15,16]. However, research has also identified transcriptional repression effects associated with abnormally high SPI1 expression, which can act as an oncogenic factor[17,18].

Ferroptosis, a recently proposed form of cell death, relies on the accumulation of intracellular $Fe^{2+}$ and the formation of lipid peroxidation[19]. Lipid peroxidation is the production of polyunsaturated fatty acids (PUFA) as substrates on cell membranes by lipoxygenase or in a non-enzymatic manner[20]. Acyl coenzyme A synthetase long-chain family member 4 (ACSL4) is an important isoenzyme for the metabolism of PUFAs, which is aberrantly expressed in a variety of tumors causing ferroptosis resistance[21,22]. Meanwhile, lipid peroxidation can also be balanced by multiple metabolic pathways, including GPX4/GSH pathway, FSP1/CoQ10 pathway and GCH1/BH4 pathway[23–25]. Erastin and RSL3 are typical ferroptosis inducers that induce ferroptosis by inhibiting the X C- system and GPX4, respectively[26,27]. Recent studies have demonstrated that targeting glutamine dependency increases vulnerability to GPX4-dependent ferroptosis in pancreatic ductal adenocarcinoma (PDAC)[28]. The activation of NRF2 metabolically reprograms and enhances pathways involved in glutamine

[1]Department of Urology, Shengjing Hospital of China Medical University, Shenyang, Liaoning, China. [2]Department of Obstetrics and Gynecology, Shengjing Hospital of China Medical University, Shenyang, Liaoning, China. [3]These authors contributed equally: Wei Xue, Zhengqi Wu. ✉e-mail: 13804016353@139.com; 13339406716@163.com

metabolism, thereby increasing ferroptosis resistance and chemotherapy resistance in pancreatic cancer. Consequently, glutaminase inhibitors can enhance the drug sensitivity of Gemcitabine[29]. Growing evidence suggests a strong association between ferroptosis and cancer in terms of development, progression, metastasis, drug resistance, and immune evasion[30], However, the research on ferroptosis in ccRCC is currently limited. Identifying vulnerable targets within the ferroptosis pathway could offer a new therapeutic strategy for treating renal cancer.

In our present study, we found that the transcription factor SPI1 is highly expressed in ccRCC and is associated with poor prognosis. Additionally, knockdown of SPI1 in combination with the ferroptosis inducer erastin significantly inhibits renal tumor cell activity. Knockdown of SPI1 induces ferroptosis in ccRCC may be caused by restoring the expression of ACSL4, a downstream target of SPI1 within renal cancer. Mechanistically, SPI1 synergises with EZH2/H3K27me3 to act on the ACSL4 promoter region to exert transcriptional repression. Therefore, we utilised an EZH2 inhibitor in combination with erastin to promote ferroptosis in ccRCC. These findings may provide a new proof-of-concept for a novel combination therapy for ccRCC targeting epigenetic factors.

## Results

### SPI1 expression is upregulated in ccRCC and correlates with poor prognosis

Given the critical role of transcription factors in tumorigenesis, progression, and metastasis[31], our study aimed to explore their impact on ccRCC. By analyzing the TCGA-KIRC database and intersecting it with the ENCODE database transcription factor dataset, we identified 11 differentially highly expressed genes (Fig. S1A, Supplementary Data 3). Literature review highlighted SPI1 as a potential key player in kidney cancer development. Analysis of TCGA data revealed significant overexpression of SPI1 not only in ccRCC (Fig. 1A) but also in various other tumors such as ESCA, GBM, KIRP, OV, and PAAD (Fig. S1B). To further explore the expression of SPI1 in ccRCC, paired tissue samples from renal cancers were subjected to qRT-PCR, aligning with TCGA data (Fig. 1B). Moreover, Immunohistochemistry (IHC) was used to detect the expression of SPI1 in normal and ccRCC tissues from 30 patients (Fig. 1C, D). We also performed western blot experiments on four renal cancer tumors to analyse the differential high expression of SPI1 (Fig. 1E). Subsequently, we found that both mRNA and protein were higher in renal cancer cell lines (Caki-1, 786-O, ACHN, A498) than in normal cells (HK-2) (Fig. 1F, G). We created a survival curve of the predicted SPI1 using The Cancer Genome Atlas (TCGA) database, which indicates that a high level of SPI1 expression was significantly correlated with shorter OS, PFS and DSS in ccRCC patients (Fig. 1H and S1C, S1D). Finally, we found that SPI1 expression was significantly correlated with the course of KIRC progression, including clinical TNM stage, invasion depth (T stage) and histological grade (Fig. 1I–K). Similarly, Western blot again verifies that SPI1 increases with increasing Fuhrman grade (Fig. 1L). Taken together, these results suggested that SPI1 is expressed at high levels in ccRCC and is associated with poor prognosis, which may serve as a predictive biomarker for ccRCC.

### Knockdown of SPI1 promotes ferroptosis in ccRCC

To further determine the function of SPI1 in ccRCC, we constructed a small interfering RNA for SPI1. The si-SPI1, which has a high knockdown efficiency, was selected for the subsequent experiments (Fig. 2A, B). It has been reported that knockdown of SPI1 to promote apoptosis in leukaemia[9,32]. However, we found that apoptosis-associated protein changes were not significant after knockdown of SPI1 in ccRCC (Fig. 2C). Interestingly, we found that SPI1 expression was negatively correlated with many polyunsaturated fatty acids, which may relate to ferroptosis (Fig. S1E). Subsequently, 786-O and ACHN cell lines were treated with the ferroptosis inducer erastin and the corresponding IC50 values were measured (Fig. 2D). CCK-8 experiments verified that si-SPI1 combined with erastin significantly inhibited tumor cell activity, which could be reverted by the ferroptosis inhibitor Ferrostatin-1 (Fer-1) (Fig. 2E). Flow cytometry (FCM)

demonstrates that si-SPI1 concomitant with erastin further promotes lipid peroxidation levels in tumor cells (Fig. 2F, G). Additionally, we observed elevated levels of MDA and $Fe^{2+}$ accumulation with si-SPI1 and erastin treatment (Fig. 2H, I). After knocking down SPI1, the electron microscopy confirmed the increased density of tumor mitochondrial membranes and the reduction or disappearance of mitochondrial cristae (Fig. 2J). In short, our results suggest that knockdown of SPI1 promotes the ferroptosis in ccRCC.

### SPI1 inhibits ferroptosis by repressing ACSL4 targets

To further investigate SPI1-promoting ferroptosis targets after knockdown of SPI1, we constructed SPI1 stable knockdown cell line by transfection of SPI1-silencing plasmids (Fig. 3A). Given that SPI1 functions as a classical transcription factor, qRT-PCR experiments were conducted to assess changes in ferroptosis-related marker genes following SPI1 knockdown, revealing a significant increase in ACSL4 levels in 786-O and ACHN cells (Fig. 3B). Subsequent western blot analysis confirmed elevated ACSL4 protein levels in SPI1 knockdown cells, consistent with the observed mRNA upregulation (Fig. 3C). The above suggests that transcription factor SPI1 may transcriptionally regulate ACSL4 expression in ccRCC. To further investigate the regulatory role of SPI1 on ACSL4, we conducted an analysis of the ACSL4 gene promoter sequence, encompassing 2 kbp upstream of the transcription start site (Fig. 3D). Within this sequence, we identified two SPI1 binding sites, namely BS1 (from −1893bp to −1917bp) and BS2 (from −1938bp to −1962bp). Dual-luciferase reporter gene analysis revealed that overexpression of SPI1 led to a significant reduction in BS1 fluorescence activity, while BS2 fluorescence activity remained largely unchanged (Fig. 3E). To determine the ability of SPI1 to bind directly to the ACSL4 promoter region, chromatin immunoprecipitation (ChIP) assays were performed, which showed that shSPI1 reduced binding to the ACSL4 promoter (Fig. 3F). To further validate that SPI1 affects ferroptosis through ACSL4, we constructed small interfering RNA sequences of ACSL4 and selected si-ACSL4 with higher knockdown efficiency for subsequent studies (Fig. S2A). CCK-8 experiments analysed that si-ACSL4 rescues ferroptosis induced due to knockdown SPI1 combined with erastin (Fig. 3G). FCM validated that si-ACSL4 rescues the increased ROS level induced due to knockdown SPI1 combined with erastin (Fig. 3H, I). Similarly, MDA level and $Fe^{2+}$ accumulation due to knockdown of SPI1 combined with erastin was also downregulated with si-ACSL4 (Fig. 3J, K). The above suggests that SPI1 transcriptionally represses ACSL4 expression, thereby inhibiting ferroptosis in ccRCC.

### Knockdown of SPI1 combined with erastin promotes ferroptosis in a ccRCC xenograft mouse model

The effect of SPI1 on ferroptosis in ccRCC development was further demonstrated in vivo based on a xenograft mouse model. The 786-O cell line of the respective treatment conditions was injected subcutaneously into the abdomen of nude mice. The tumors were measured once every six days. Mice were then sacrificed to measure tumor weight and volume (Fig. 4A). Compared to the DMSO group, the volume and weight of tumors were significantly reduced in the shSPI1 combined with erastin groups, which could be reversed by si-ACSL4 (Fig. 4B–D). Notably, the mRNA and protein expression levels of SPI1 and ACSL4 in xenograft tumors were examined by qRT–PCR and western blot, which again confirmed the regulation of ACSL4 expression by SPI1 in vivo (Fig. 4E, F). Moreover, shSPI1 combined with erastin can also promote lipid peroxidation levels in vivo (Fig. 4G, H). Therefore, these in vivo experiments demonstrated that knockdown of SPI1 combined with erastin promotes ferroptosis.

### SPI1 inhibits ACSL4 expression by recruiting EZH2-mediated H3K27me3

SPI1 does not contain any transcriptional repressor structure with, but may exert transcriptional repression by interacting with some co-transcription factors[33]. In general, HDAC1 acts as a synergistic mediator of SPI1-induced transcriptional repression, or when SPI1 is located in the enhancer

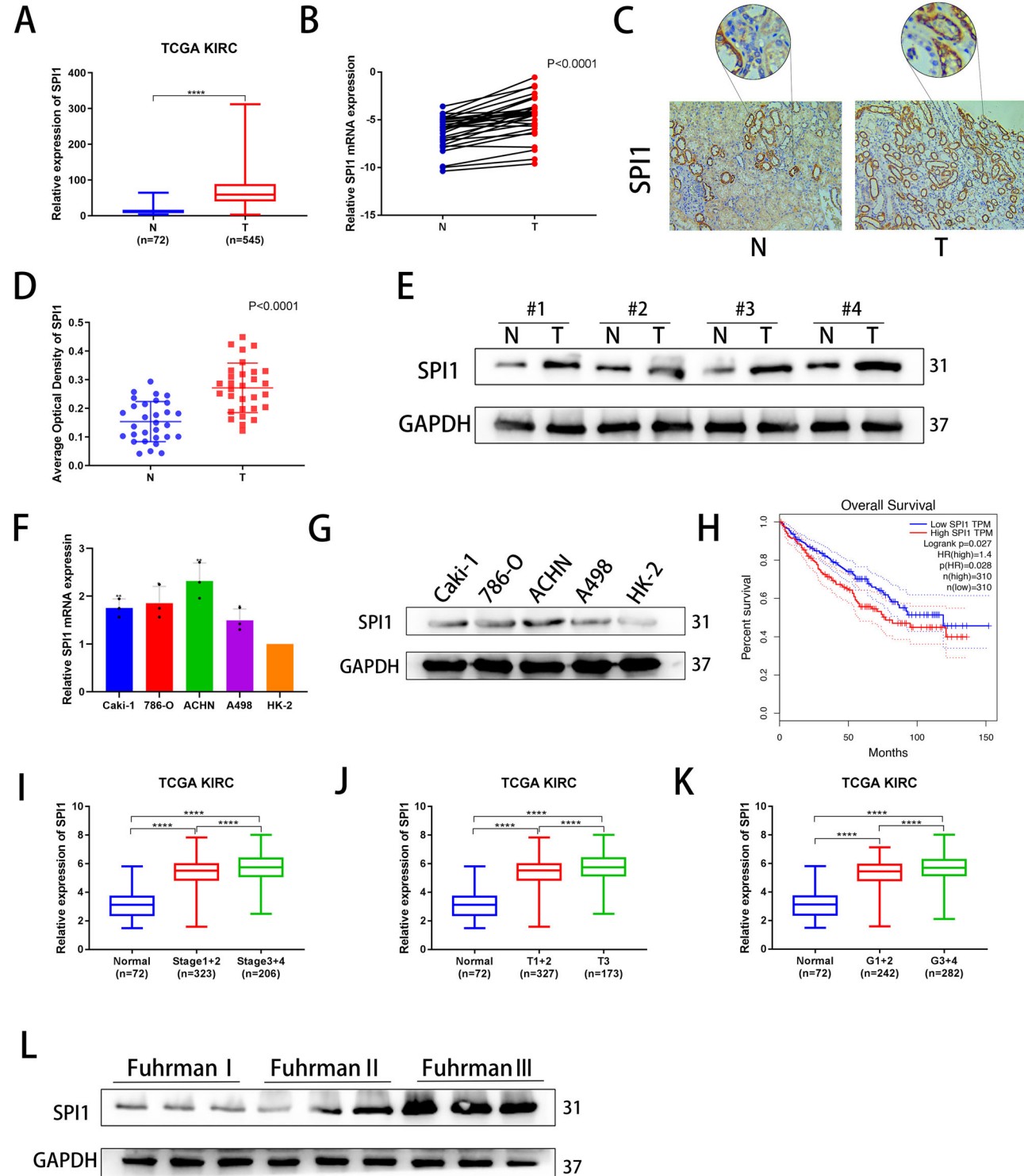

**Fig. 1 | SPI1 expression is upregulated in ccRCC and correlates with poor prognosis. A** The mRNA expression of SPI1 in normal renal tissues and renal cancer tissues from TCGA-KIRC database. **B** SPI1 mRNA expression in 30 pairs of ccRCC samples (n = 30). **C, D** Immunohistochemical staining examined the protein expression of SPI1 in 30 tumor tissues and adjacent nontumor tissues (n = 30). **E** Western blot analysis showed SPI1 expression in 4 pairs of renal tumors and their corresponding paraneoplastic samples (n = 4). **F, G** qRT-PCR and Western blot verification of SPI1 mRNA and protein levels in Caki-1, 786-O, ACHN A498 and HK-2 cell lines. **H** Overall survival (OS) Kaplan–Meier curve for SPI1 based on TCGA-KIRC. **I-K** Correlation analysis of SPI1 mRNA expression and histological grade, clinical TNM stage, as well as depth of invasion n (T stage) based on TCGA. **L** Western blot analysis showed that the expression of SPI1 in different Fuhrman grade. Data are presented as the mean ± SD and analyzed by t-test or Kruskal–Wallis test. *: P < 0.05, **: P < 0.01, ***: P < 0.001.

sequence, polycomb suppression complex 2 (PRC2) enhances gene transcriptional repression by depositing H3K27me3 on the promoter sequence[34]. In order to find the mechanism of SPI1 transcriptional repression, we treated tumor cells with HDAC inhibitor (Entinostat) and PCR2

inhibitor (UNC1999) for a Western blot assay. The results demonstrated that UNC1999 significantly increased ACSL4 expression, whereas Entinostat had no significant effect (Fig. 5A and S2B). In addition, treatment with UNC1999 resulted in a significant increase in ACSL4 protein levels with

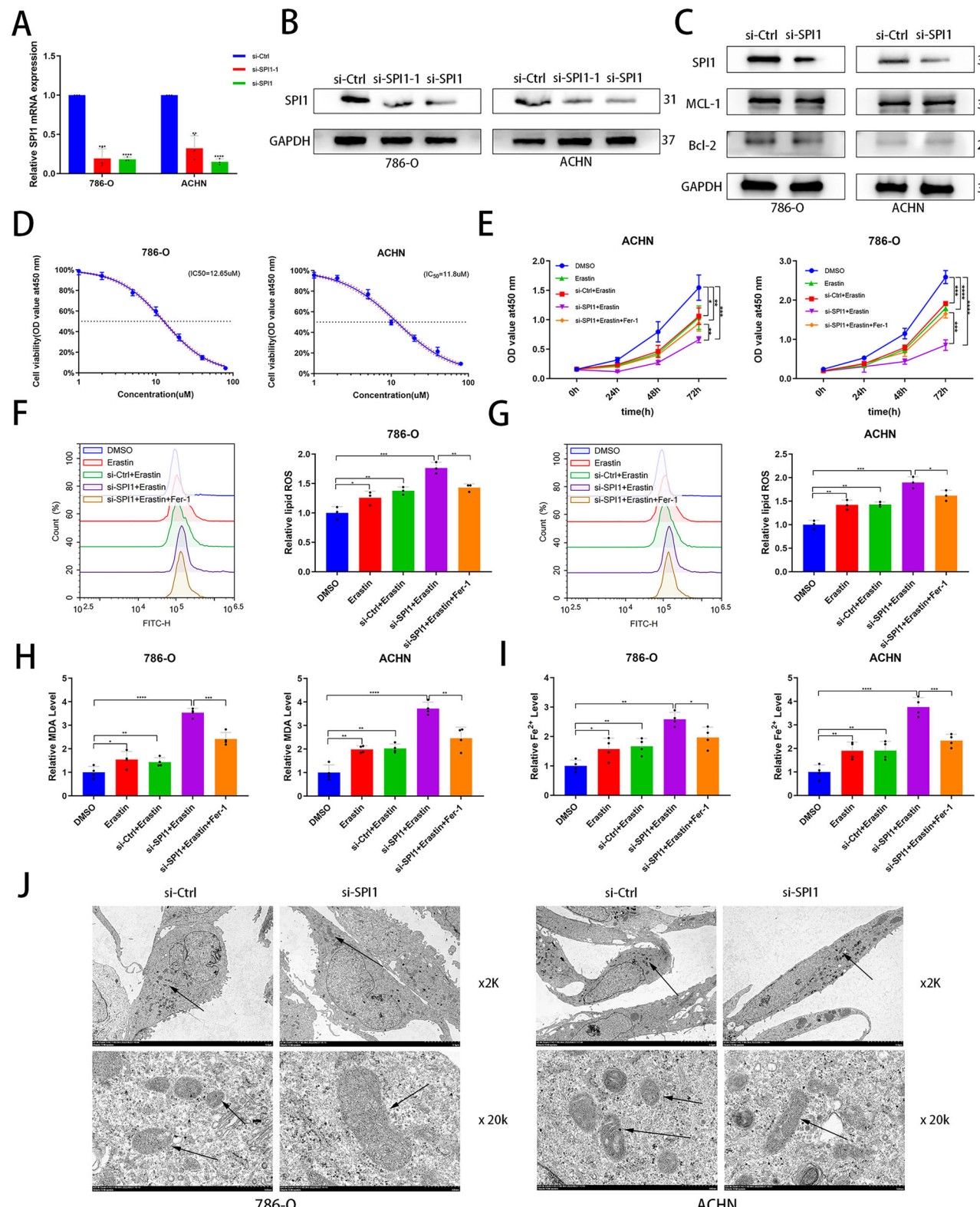

**Fig. 2 | Knockdown of SPI1 promotes ferroptosis in ccRCC. A**, **B** qRT-PCR and western blot examined the efficiency of SPI1 knockdown in 786-O and ACHN. **C** SPI1, MCL-1 and Bcl-2 proteins were detected in the si-Ctrl group and si-SPI1 group of 786-O and ACHN cell lines. **D** IC50 values of the corresponding cell lines measured after treatment of 786-O and ACHN with erastin. **E** CCK-8 assays were used to analyse the effect on cell viability in 786-O and ACHN. **F**, **G** Flow cytometry was used to detect lipid peroxidation level in 786-O and ACHN (n = 3). **H**, **I** MDA and Fe$^{2+}$ levels were measured separately in 786-O and ACHN (n = 4). **J** Electron microscopy imaging was conducted in 786-O and ACHN. Mitochondria are marked by black arrows. Data are presented as the mean ± SD and analyzed by t-test or Kruskal-Wallis test. *: P < 0.05, **: P < 0.01, ***: P < 0.001.

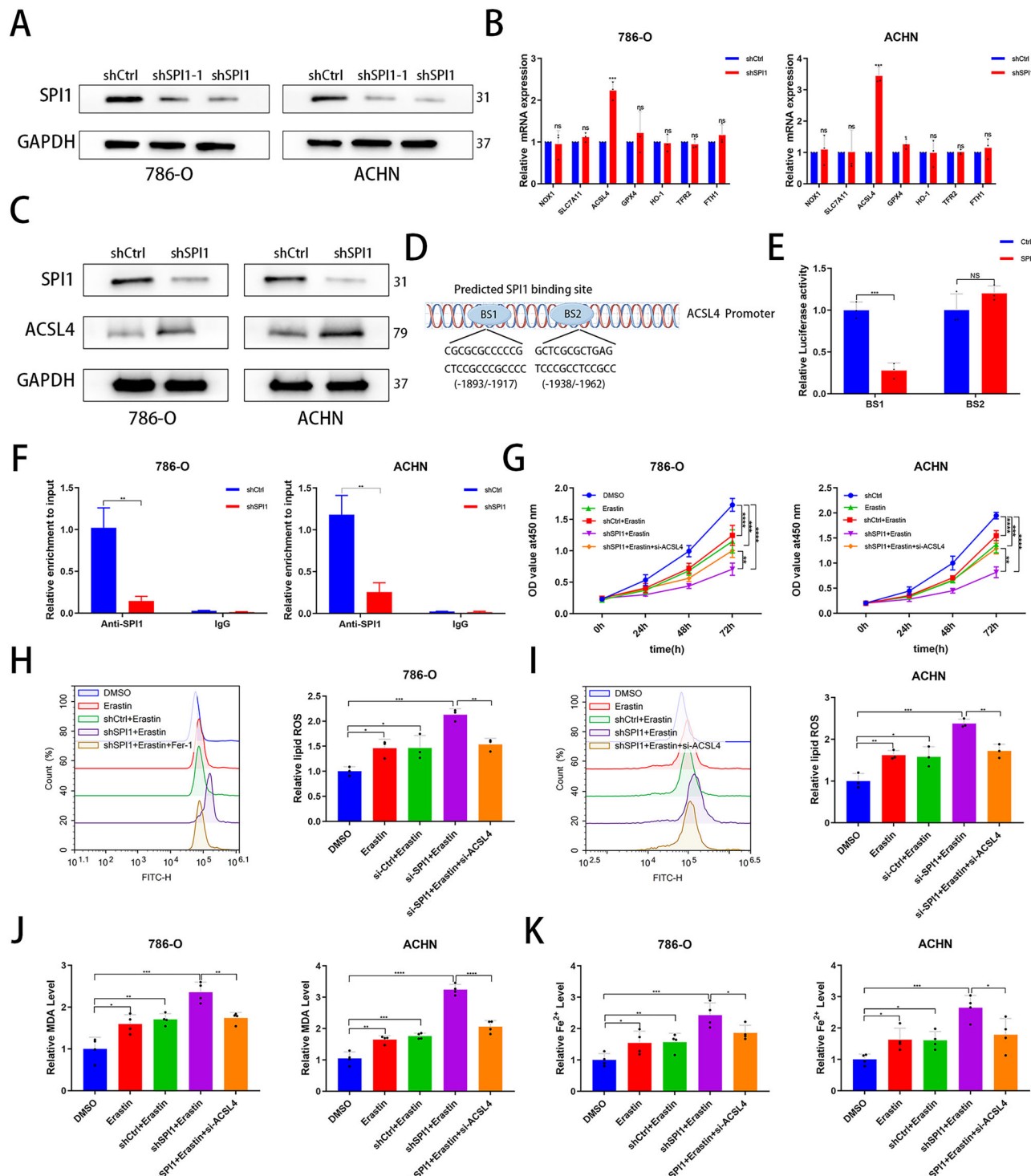

**Fig. 3 | SPI1 inhibits ferroptosis by repressing ACSL4 targets. A** Western blot examined the efficiency of SPI1 knockdown transfected with the shSPI1 in 786-O and ACHN. **B** qRT-PCR detection of NOX1, SLC7A11, ACSL4, GPX4, HO-1, NRF2 and FTH1 mRNA expression in shCtrl and shSPI1 groups (n = 3). **C** SPI1and ACSL4 proteins were detected in the si-Ctrl group and si-SPI1 group of 786-O and ACHN cell lines. **D** It is predicted that SPI1 may bind to the BS1 and BS2 sites of the 2000 bp sequence upstream of the ACSL4 transcription start site according to the HumanTFDB. **E** The dual-luciferase reporter assay showed the activity of the ACSL4 promoter fragment in 293 T cell line (n = 3). **F** ChIP-PCR analysis of the binding of SPI1 to the promoter of ACSL4 in 786-O and ACHN cell lines (n = 3). **G** CCK-8 assays were used to analyse the effect on cell viability in 786-O and ACHN. **H, I** Flow cytometry was used to detect lipid peroxidation level in 786-O and ACHN (n = 3). **J, K** MDA and Fe$^{2+}$ levels were measured separately in 786-O and ACHN (n = 4). Data are presented as the mean ± SD and analyzed by t -test or Kruskal-Wallis test. *: P < 0.05, **: P < 0.01, ***: P < 0.001.

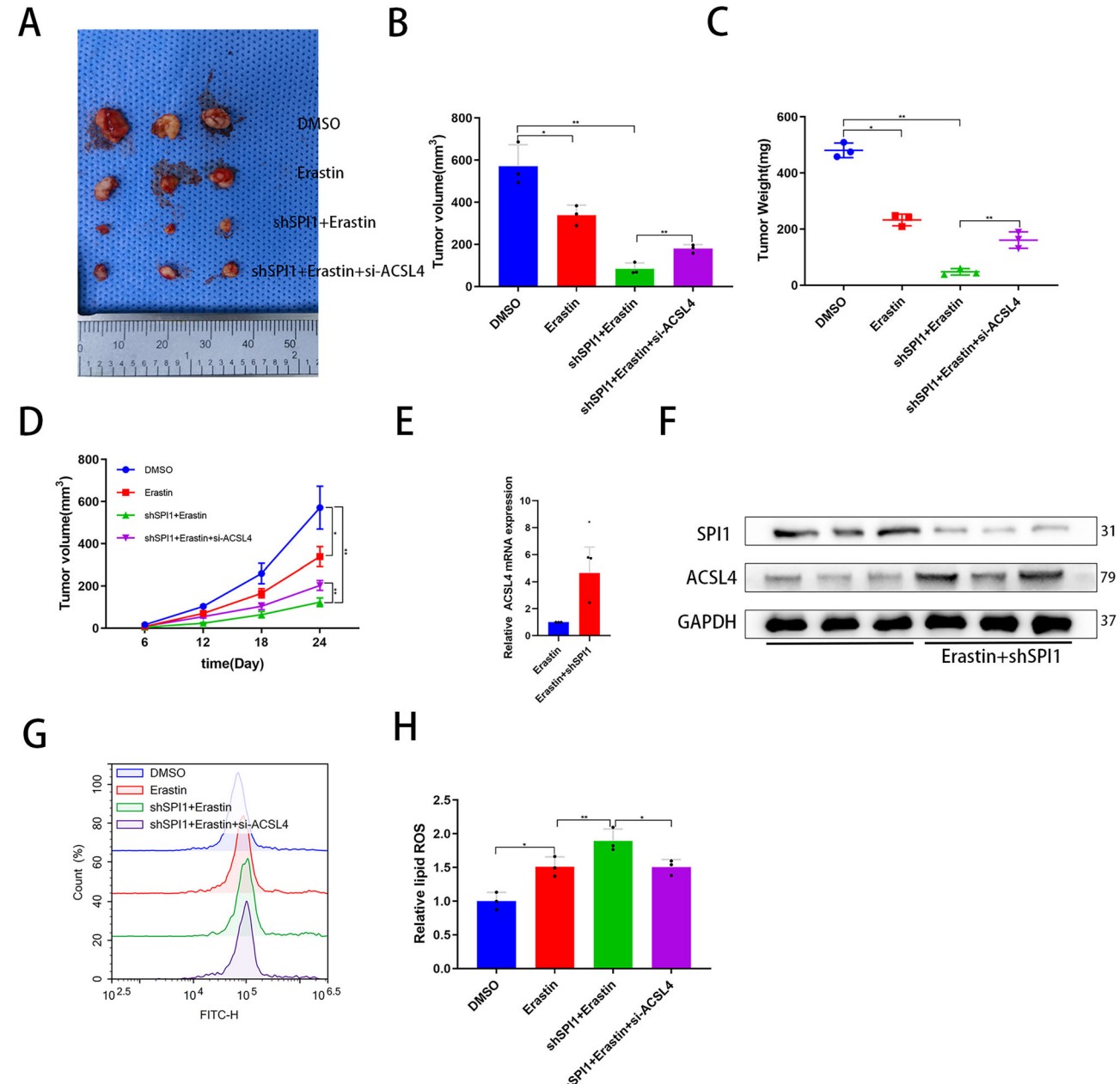

**Fig. 4 | Knockdown of SPI1 combined with erastin promotes ferroptosis in a ccRCC xenograft mouse model. A** Subcutaneous tumors of various treatment conditions were demonstrated. **B, C** The final tumor volume and weight were measured at 26-day. **D** The tumor volume of various treatment conditions was measured at 6, 12, 18 and 24 days. **E** qRT-PCR for mRNA expression of ACSL4 in tumors of nude mice treated with the corresponding conditions (n = 3). **F** Western bolt for protein expression of ACSL4 in tumors of nude mice treated with the corresponding conditions (n = 3). **G, H** Flow cytometry was used to detect lipid peroxidation level in the DMSO, Erastin, shSPI1+Erastin as well as shSPI1+Erastin +si-ACSL4 groups in tumors of nude mice (n = 3). Data are presented as the mean ± SD and analyzed by t -test or Kruskal-Wallis test. *: P < 0.05, **: P < 0.01, ***: P < 0.001.

increasing time (Fig. 5B). We speculate that EZH2, a core component of the PRC2 complex, is involved in the regulation of ACSL4 transcriptional repression by SPI1. The TCGA-KIRC database shows that EZH2 is highly expressed and is associated with poor prognosis (Fig. S2C, D). Moreover, SPI1 level was positively correlated with EZH2 level in ccRCC (Fig. S2E). Subsequently, ACSL4 mRNA and protein levels were increased using qRT-PCR and western blot detection after knockdown of EZH2 (Fig. 5C, D). We constructed an EZH2 overexpression plasmid for transfection into renal cancer cell lines and verified the EZH2 overexpression efficiency (Fig. S2F, G). Overexpressing EZH2 partially reversed the promotional effect of SPI1 knockdown on ACSL4 protein levels (Fig. 5E). These findings indicate that EZH2-mediated H3K27me3 is also involved in the transcriptional

repression of ACSL4 by SPI1. The HumanTFDB website was used to predict the potential binding sites for EZH2 in the promoter of ACSL4 (Fig. 5F, Supplementary Data 2). Correspondingly, ChIP-seq data in UCSC indicate that H3K27me3 also has a binding site in the ACSL4 promoter region (Fig. S2H). To verify that EZH2 interacts with the ACSL4 promoter region, we performed ChIP-PCR assay using EZH2 or H3K27me3 antibody in the selected region of ACSL4 promoter as prediction (Fig. 5G). In addition, immunofluorescence analysis showed that SPI1 and EZH2 co-localize substantially (Fig. 5H). Subsequent co-IP experiments verified that SPI1 can interact with EZH2 in ccRCC (Fig. 5I), suggesting that SPI1 may recruit EZH2 and consequently induce transcriptional repression. Finally, to verify that SPI1 inhibits ACSL4 expression through EZH2-H3K27me3, we

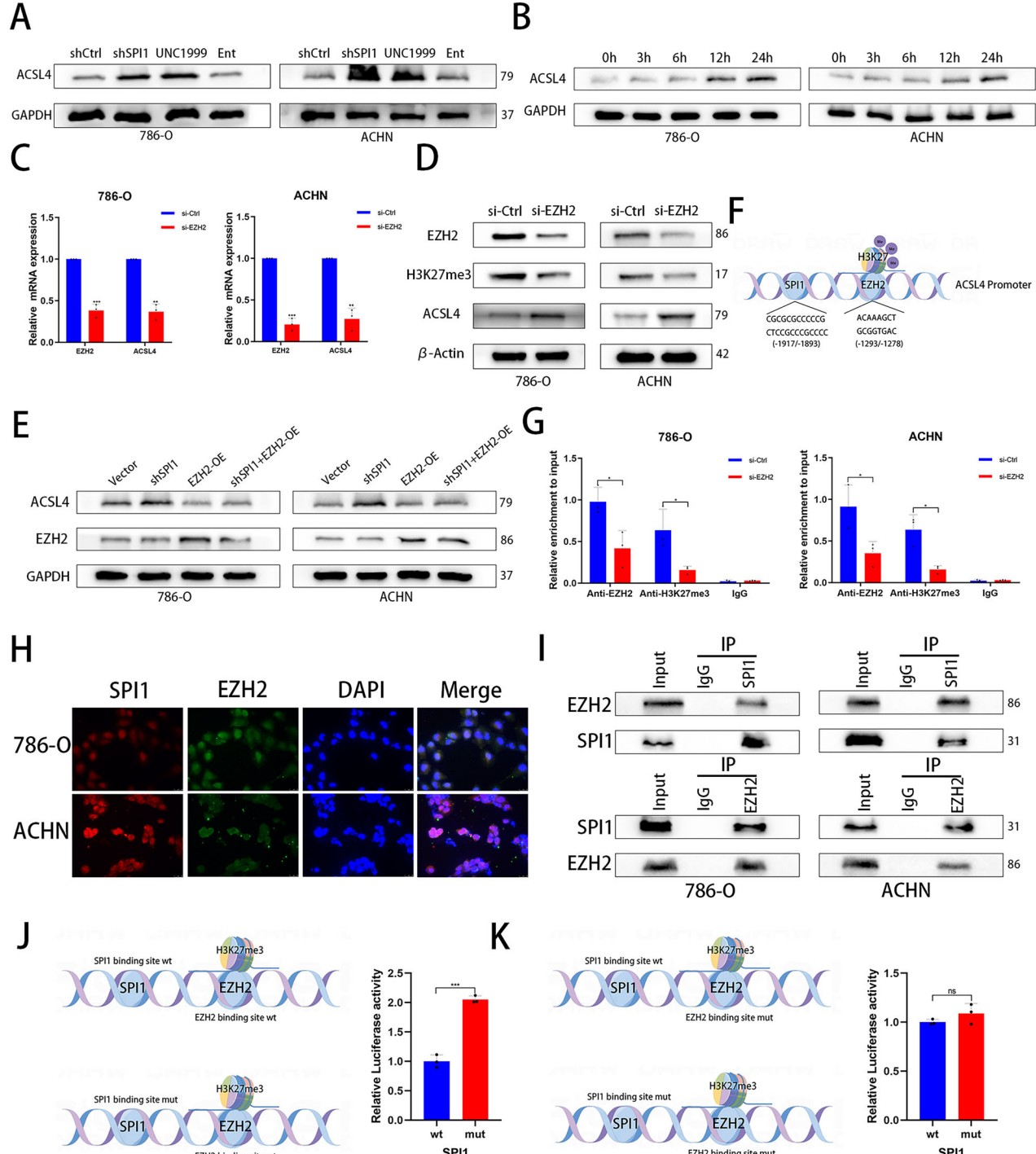

**Fig. 5 | SPI1 inhibits ACSL4 expression by recruiting EZH2-mediated H3K27me3.** **A** Western blot analysis of ACSL4 protein in 786-O and ACHN cells treated with shCtrl, shSPI1, UNC1999 (5 μM) and Entinostat (300 nM). **B** Western blot analysis of ACSL4 protein expression levels with different time gradients of 5uM concentration of UNC1999 treated 786-O and ACHN cells. **C** qRT-PCR for mRNA expression of EZH2 and ACSL4 in si-Ctrl and si-EZH2 groups (n = 3). **D** Western bolt for protein expression of EZH2, H3K27me3 and ACSL4 in si-Ctrl and si-EZH2 groups. **E** Western bolt for protein expression of EZH2 and ACSL4 in 786-O and ACHN. **F** Prediction of EZH2 binding site upstream of ACSL4 promoter based on HumanTFDB. **G** ChIP-PCR analysis of the binding of EZH2 and H3K27me3 to the promoter of ACSL4 in 786-O and ACHN. **H** Immunofluorescence staining of SPI1 (red), EZH2 (green) and DAPI (blue) (n = 3). **I** Lysates from 786-O and ACHN cells were immunoprecipitated (Ip) with an anti-SPI1 antibody or immunoglobulin G (IgG) as indicated. Immunoprecipitates were analysed by immunoblotting with anti-EZH2 and anti-SPI1 antibodies. **J, K** Schematic representation of the ACSL4 promoter–luciferase reporter constructs. Two binding sites of the ACSL4 promoter (wild type or mutant) were cloned into the pGL4 reporter vector. The SPI1 binding site and EZH2 binding site are indicated (n = 3). Data are presented as the mean ± SD and analyzed by t -test or Kruskal-Wallis test. *: P < 0.05, **: P < 0.01, ***: P < 0.001.

constructed plasmids containing the ACSL4 promoter sequence in the 293 T cell line (The SPI1 and EZH2 binding site is wild-type or mutant) (Fig. 5J, K). Luciferase reporter gene assay showed that overexpressing SPI1 failed to inhibit ACSL4 expression after mutation of the EZH2 binding site (Fig. 5J, K). In conclusion, our results suggest that SPI1 inhibits ACSL4 expression through EZH2-H3K27me3 pathway.

### EZH2 overexpression reversed the ferroptosis-related phenotype reversed induced by SPI1 knockdown

To further validate that SPI1 affects the ferroptosis-related phenotype by recruiting EZH2, we performed CCK-8 experiments to show that over-expression EZH2 partially restored cellular activity due to SPI1 knockdown in 786-O and ACHN (Fig. 6A). Similarly, we found that EZH2 over-expression partially reverted the accumulation of $Fe^{2+}$ and MDA changes produced due to SPI1 knockdown in 786-O and ACHN (Fig. 6B, C). The results of FCM and IF showed that EZH2 overexpression partially reversed the down-regulation of lipid peroxidation due to SPI1 knockdown (Fig. 6D–F). In conclusion, our experimental results suggest that SPI1 affects ferroptosis-related phenotypes through EZH2.

### The EZH2 inhibitor UNC1999 combined with erastin induces ferroptosis and inhibits tumor growth in ccRCC

Previous studies have shown that shSPI1 promotes ferroptosis through EZH2. Therefore, we investigated the potential inhibitory effects of combining the EZH2 inhibitors UNC1999 and erastin on tumors. Our CCK-8 assay revealed that the combination of UNC1999 and erastin significantly reduced cellular activity, which was reversed by Fer-1 (Fig. 7A). Additionally, we observed that the combination of UNC1999 and erastin led to increased $Fe^{2+}$ accumulation, MDA accumulation, and lipid peroxidation levels, all of which were reversed by Fer-1 in 786-O and ACHN cell lines (Fig. 7B–E). Subsequently, we utilized a nude mouse cell-derived xenograft (CDX) model to assess the impact of this combination on tumor growth in vivo. As anticipated, UNC1999 combined with erastin resulted in a more pronounced inhibition of tumor size, volume, and weight (Figs. 7F and S3E, F). RNA was extracted for qRT-PCR and protein for WB analysis from nude tumor samples. Surprisingly, UNC1999 was found to enhance the expression of ACSL4 mRNA and protein, indirectly confirming the inhibition of ACSL4 expression by EZH2 in an in vivo setting (Fig. 7G, H). To further validate that UNC1999 induce ferroptosis through ACSL4 targets, we knocked down ACSL4 for ferroptosis-related phenotypic revert. The CCK-8 assay revealed that knockdown ACSL4 partially reversed tumor cell death caused by UNC1999 combined with erastin in 786-O and ACHN (Fig. 7I and S3A). Similarly, we again detected that ACSL4 knockdown partially reversed the accumulation of $Fe^{2+}$ and MDA, and increased levels of lipid peroxidation due to UNC1999 combined with erastin treatment (Fig. 7J–L, S3B–S3D). In summary, treatment with the EZH2 inhibitor UNC1999 in combination with the ferroptosis inducer erastin promotes tumor ferroptosis in vivo and in vitro.

### ACSL4 is correlated with SPI1 in ccRCC tissue samples

Previous experiments have confirmed that SPI1 transcriptionally represses ACSL4 expression. We aimed to explore the relationship between SPI1 and ACSL4 in kidney cancer tissue samples. Our analysis revealed that ACSL4 showed significantly low expression levels in the TCGA-KIRC database (Fig. 8A), while SPI1 was highly expressed differentially (Fig. 1A). Meanwhile, we also analysed that low ACSL4 expression was associated with poor prognosis and shortened OS and PFS survival (Figs. 8B and S3G). The expression and prognosis of ACSL4 were completely opposite to that of SPI1, implying that the low expression of ACSL4 may be partially regulated by SPI1 in ccRCC. The correlation between SPI1 and ACSL4 was studied in renal cancer cell lines and tissue samples. Analysis using qRT-PCR and IHC showed a negative correlation between the mRNA and protein levels of SPI1 and ACSL4 in 30 tissue samples (Fig. 8C, D). Further examination of expression levels in various cell lines confirmed a negative correlation

between SPI1 and ACSL4 (Fig. 8E). Additionally, analysis of 8 pairs of ccRCC cancer tissues and corresponding adjacent tissues revealed a negative correlation in SPI1 and ACSL4 expression levels (Fig. 8F). Finally, IF assay was performed in SPI1-depleted and non-depleted cancer cells and found that ACSL4 expression levels increased after SPI1 depletion (Fig. 8G) In Summary, the above experiments illustrate that SPI1 negatively correlates with ACSL4 in ccRCC cell line and tissue samples.

## Discussion

ccRCC is the most common subtype of renal cell carcinoma, which can be cured by surgery in its early stages[1]. Unfortunately, due to the lack of noticeable symptoms during the initial phases of ccRCC, approximately 35% of patients miss the opportunity for surgical intervention and are diagnosed at a later stage[35]. Because kidney cancer is not sensitive to radiotherapy, immunotherapy and targeted therapy will become a new trend in the treatment of advanced kidney cancer[5,36]. Therefore, our goal is to find more effective therapeutic targets and prognostic markers for renal cancer to solve the real problems.

In our study, SPI1 was found to be differentially overexpressed in ccRCC and associated with poor clinical prognosis. In both in vivo and in vitro experiments, knocking down SPI1 in combination with erastin led to the promotion of ferroptosis. Knockdown of SPI1 resulted in the upregulation of ACSL4, a target associated with ferroptosis, leading to increased lipid peroxidation. Simultaneously, the introduction of erastin, a ferroptosis inducer, disrupted the reducing system, further enhancing ferroptosis in tumor cells (Fig. 8G). Mechanistically, SPI1 collaborates with EZH2 to facilitate H3K27me3 modification, influencing the transcription of ACSL4. Ultimately, we employed the EZH2 inhibitor UNC1999 in conjunction with erastin to induce ferroptosis in ccRCC. Our experimental results demonstrate the enrichment of SPI1 and EZH2 in the promoter region of ACSL4, indicating that SPI1 may be involved in the epigenetic silencing of ACSL4 through the recruitment of EZH2. However, the current data cannot definitively determine whether SPI1 directly recruits EZH2 to the chromatin or if this co-localization is mediated by an intermediary protein complex, which requires further exploration through related experiments in the future.

The SPI1 gene encodes the transcription factor PU.1, a member of the ETS family of transcription factors (E26 translationally specific family) that was first focused on in leukaemia[37]. Recent studies have shown that SPI1 may be a crucial oncogenic transcription factor promoting the progression of various cancers, including liver cancer, gastric cancer and ovarian cancer[38–40]. Despite lacking a specific transcriptional repressor sequence, SPI1 can interact with transcriptional co-repressors to exert a repressive effect[33,34]. Additionally, SPI1 synergizes with epigenetic molecules to mediate altered chromatin conformation and exert transcriptional regulation. For example, SPI1 can recruit or stabilize HDAC1, leading to inhibition of enhancer interactions or promotion of acetylation at target gene transcription start sites[34]. In previous studies, it has been noted that SPI1 interacts with histone acetyltransferase (HAT) CBP/P300[15,16], raising questions about SPI1's role in the balance between HAT and HDAC. In our study, we have verified that SPI1 suppresses the transcription of ACSL4 via EZH2-mediated H3K27me3, rather than through HDAC.

In addition, we utilized clinical sample analysis to identify differential expression of SPI1 and ACSL4 in renal cancer, consistent with the TCGA database. High differential expression of SPI1 or low differential expression of ACSL4 is significantly associated with poor prognosis. The TCGA database demonstrates that SPI1 expression is significantly associated with the progression of KIRC, including clinical TNM stage, invasion depth (T stage) and histological grade. Meanwhile, our experimental results demonstrate that when SPI1 is highly expressed or ACSL4 is lowly expressed, the combination of ferroptosis inducers with EZH2 inhibitors can promote ferroptosis in ccRCC. This indicates that the expression levels of SPI1 or ACSL4 can serve as predictive biomarkers for identifying patient populations likely to benefit from combination therapy with ferroptosis inducers and EZH2 inhibitors.

Ferroptosis is a novel form of cell death proposed in recent years, dependent on the accumulation of intracellular $Fe^{2+}$ and the formation of

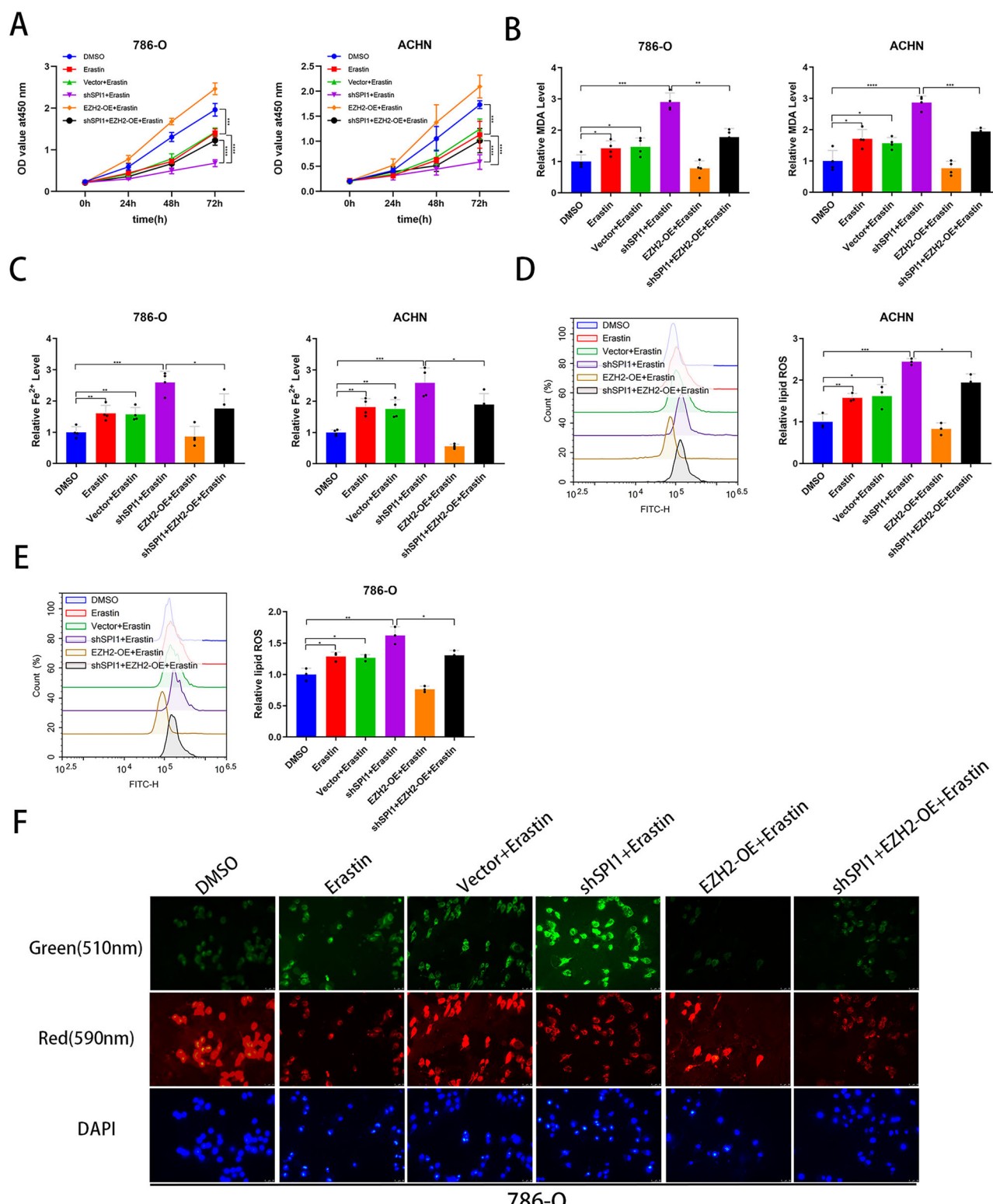

**Fig. 6 | EZH2 overexpression reversed the ferroptosis-related phenotype reversed induced by SPI1 knockdown. A** CCK-8 assays were used to analyse the effect on cell viability in 786-O and ACHN. **B, C** MDA and Fe2+ levels were measured separately in 786-O and ACHN (n = 4). **D, E** Flow cytometry was used to detect lipid peroxidation level in 786-O and ACHN (n = 3). **F** Immunofluorescence was used to detect lipid peroxidation in 786-O and ACHN, the lipid peroxidation prefers enhanced light absorption at 510 nm (green). Data are presented as the mean ± SD and analyzed by t test or Kruskal–Wallis test. *: P < 0.05, **: P < 0.01, ***: P < 0.001.

lipid peroxidation[19]. Studies have shown that abnormal lipid metabolism is a typical characteristic of renal cancer, and hypoxia-inducible factor promotes ferroptosis escape by inhibiting fatty acid oxidation[41]. Additionally, Dipeptidyl peptidase 9 (DPP9) inhibits ferroptosis through the KEAP1-NRF2 pathway, leading to resistance to sorafenib in ccRCC[42]. Literature reports that

SPI1 interacts with IRF1 or TIMP1 to regulate the sensitivity of colon cancer to ferroptosis[43,44]. While limited research has delved into the correlation between ccRCC and ferroptosis, the role of SPI1 remains ambiguous in the context of kidney cancer and ferroptosis. Our investigation revealed that SPI1 fosters resistance to ferroptosis by suppressing ACSL4, a crucial enzyme

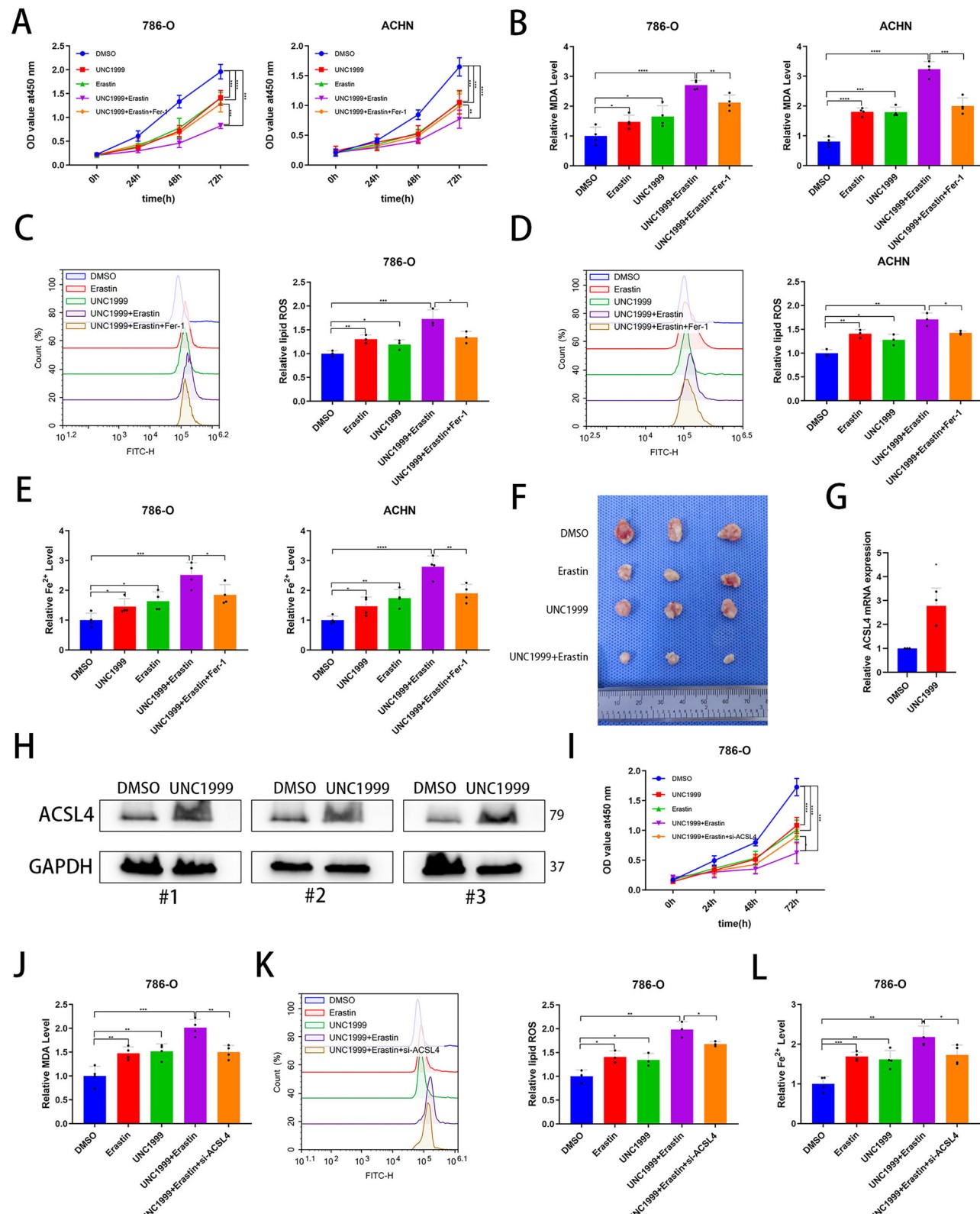

**Fig. 7 | EZH2 overexpression reversed the ferroptosis-related phenotype reversed induced by SPI1 knockdown. A** CCK-8 assays were used to analyse the effect on cell viability in 786-O and ACHN. **B** MDA levels were measured separately in 786-O and ACHN (n = 4). **C, D** Flow cytometry was used to detect lipid peroxidation level in 786-O and ACHN (n = 3). **E** Fe$^{2+}$ levels were measured separately in 786-O and ACHN (n = 4). **F** Shows subcutaneous tumors and isolated tumors from 12 nude mice under different treatment conditions. **G** qRT-PCR for mRNA expression of

ACSL4 in tumors of nude mice treated with the corresponding conditions (n = 3). **H** Western bolt for protein expression of ACSL4 in tumors of nude mice treated with the corresponding conditions (n = 3). **I** CCK-8 assays were used to analyse the effect on cell viability in 786-O. **J** MDA levels were measured separately in 786-O. **K** Flow cytometry was used to detect lipid peroxidation level in 786-O. **L** Fe2+ levels were measured separately in 786-O. Data are presented as the mean ± SD and analyzed by t test or Kruskal-Wallis test. *: P < 0.05, **: P < 0.01, ***: P < 0.001.

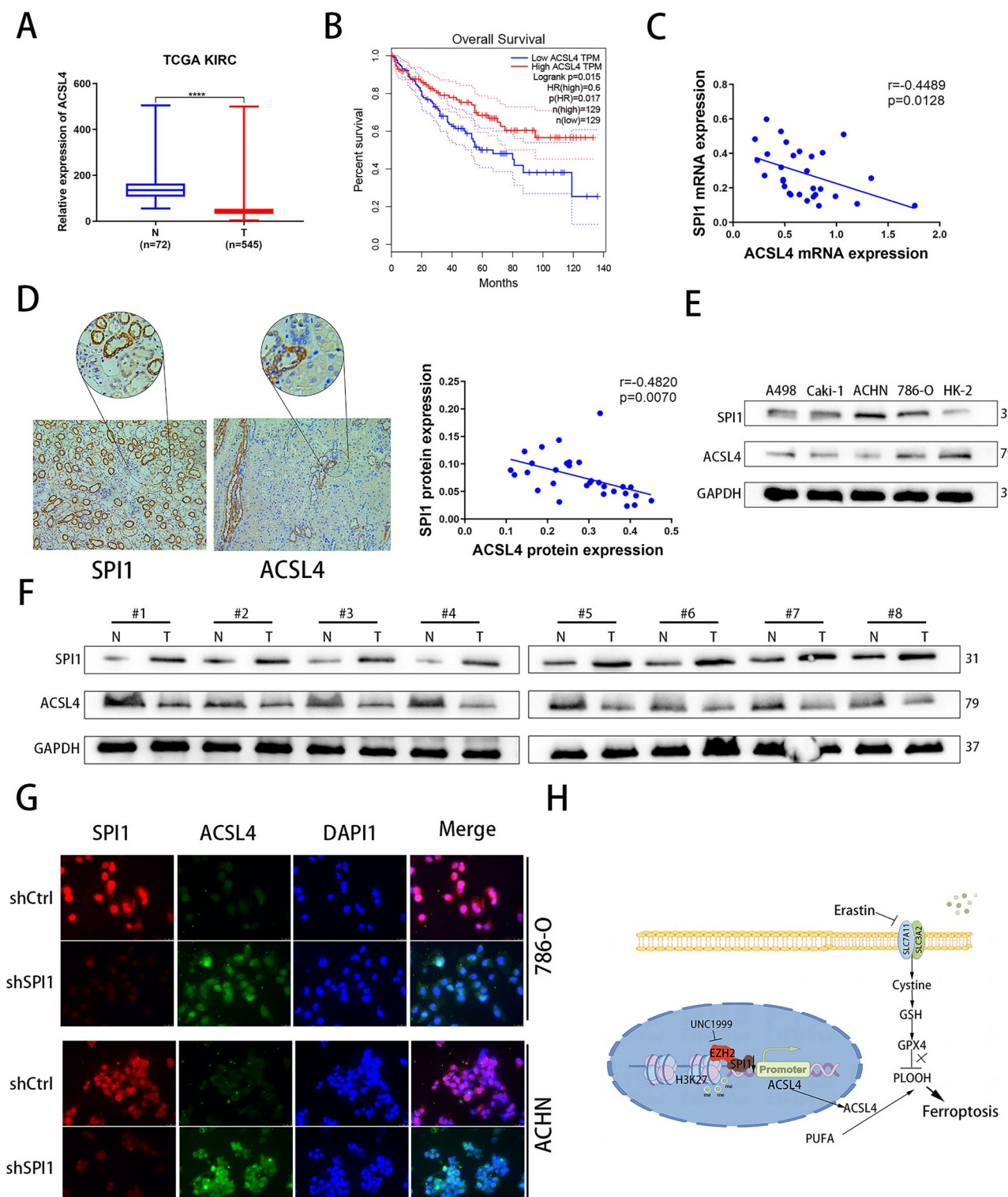

**Fig. 8 | ACSL4 is correlated with SPI1 in ccRCC tissue samples. A** The mRNA expression of ACSL4 in normal renal tissues and renal cancer tissues from TCGA-KIRC database. **B** Overall survival (OS) Kaplan–Meier curve for ACSL4 based on TCGA-KIRC. **C** qRT‒PCR was used to analyse correlation of SPI1 and ACSL4 mRNA expression in 30 pairs of ccRCC samples (n = 30). **D** IHC was used to analyse correlation of SPI1 and ACSL4 protein expression in 30 pairs of ccRCC samples (n = 30). **E** Western blot verification of SPI1 and ACSL4 protein levels in A498, Caki-1, ACHN, 786-O and HK-2 cell lines. **F** Western blot analysis for the protein levels of SPI1 and ACSL4 in 8 paired clinical samples with ccRCC and adjacent tissues (n = 8). **G** Immunofluorescence staining of SPI1 (red), ACSL4(green) and DAPI (blue) in 786-O and ACHN. **H** Article Mechanism Diagram: Knockdown of SPI1 or treatment with UNC1999 both resulted in the restoration of ACSL4 expression in renal cancer cells. Additionally, the combined application of Erastin further enhanced lipid peroxidation, leading to ferroptosis in ccRCC. Data are presented as the mean ± SD and analyzed by *t* test or Kruskal-Wallis test. *: P < 0.05, **: P < 0.01, ***: P < 0.001.

that facilitates lipid peroxidation. Since ACSL4 facilitates the integration of polyunsaturated fatty acids (PUFAs) into the phospholipids of cell membranes, it enhances sensitivity to ferroptosis. Therefore, the ferroptosis inducers erastin/RSL3 may be closely related to the expression level of ACSL4 when inducing cellular ferroptosis. Studies have indicated that tumors exhibiting high ACSL4 expression, such as certain triple-negative breast cancers and liver cancers, may demonstrate increased sensitivity to RSL3[45]. In future research, we will employ single-cell sequencing technology to analyze the differences in sensitivity to erastin/RSL3 among various renal cancer subpopulations characterized by differing ACSL4 expression levels.

Although the binding of SPI1 to its promoter region is required for ACSL4 to be regulated, it is the EZH2-mediated H3K27me3 that exerts transcriptional repression. EZH2, the enzymatically active core subunit of the PRC2 complex, methylates lysine residues of histone H3 at position 27, leading to chromosomal structural compaction and gene silencing[46]. Studies on the human cancer genome have revealed high expression of EZH2 in various cancers, with involvement in different biological functions, including ferroptosis[47]. In tongue squamous cell carcinoma, EZH2 inhibits erastin-induced ferroptosis through the miR-125b-5p/SLC7A11 pathway rather than the H3K27me3 pathway[48]. Conversely, in autoimmune diseases, EZH2 can promote ferroptosis by inhibiting SLC7A11[49]. These findings are not paradoxical, as different diseases have different molecular mechanisms and involve different molecular factors. Given that EZH2 is a driver gene for a number of cancers, the development of EZH2 inhibitors has become an active field. In our study, we chose the small molecule compound UNC1999, a dual inhibitor of EZH1 and EZH2, which showed significant efficacy against leukaemia, bladder cancer and colon cancer[50–52]. The combined application of UNC1999 and erastin has demonstrated promising antitumor efficacy in the ccRCC model; however, potential off-target or systemic toxicity remains a challenge that we need to address. Nanoparticles can passively target tumors through the enhanced permeability and retention (EPR) effect or actively target them via surface modifications, such as folate and transferrin receptor ligands[53,54]. Liposomes can prolong the half-life of drugs and reduce non-specific distribution, which significantly reduces drug toxicity[55,56]. The new generation of ADCs, such as Enhertu (DS-8201), enhances efficacy and reduces toxicity through cleavable linkers and a high drug-to-antibody ratio (DAR)[57]. Additionally, localized drug delivery strategies are crucial for mitigating drug toxicity. For instance, intratumoral injection of PD-1 inhibitors combined with oncolytic viruses enhances anti-tumor immune responses[58], and intratumoral injection of the oncolytic virus T-VEC (Imlygic) is used for melanoma treatment[59]. Injectable hydrogels continuously release drugs, such as doxorubicin, within the postoperative tumor cavity, thereby minimizing systemic diffusion[60]. Enzyme-activated prodrugs are also a key approach to addressing drug toxicity reactions. γ-Glutamyltranspeptidase (GGT)-activated prodrugs release active drugs in regions with high enzyme expression in tumors, thereby targeting tumor cells[61]. Some prodrugs, such as Tirapazamine derivatives, are activated in hypoxic or high reactive oxygen species (ROS) environments within tumors[62]. Therefore, we will conduct in vivo toxicity response-related studies in subsequent research to enhance the clinical feasibility of the proposed therapeutic strategy.

In conclusion, we found that SPI1 is highly expressed in ccRCC and is associated with poor prognosis. Furthermore, we validated that knockdown of SPI1 combined with treatment erastin promotes renal cancer ferroptosis, elaborating on the mechanism that SPI1 interacts with EZH2 to mediate the transcriptional repression of ACSL4 targets by H3K27me3. The experiments validate the efficacy of UNC1999 in combination with erastin in inhibiting ccRCC both in vivo and in vitro.

## Methods and materials
### Cell lines, cell culture and tissue samples
Human normal renal cell lines HK-2 and renal cancer cell lines (786-O, ACHN, Caki-1, A498) were procured from the Cell Resources Center at the Shanghai Academy of Life Sciences, Chinese Academy of Sciences. The renal cell lines were maintained in the appropriate culture medium supplemented with 10% heat-inactivated fetal bovine serum (Thermo Fisher Scientific) and 1% streptomycin/penicillin (Keygen, Nanjing, China) at 37 °C and 5% $CO_2$. Tissue samples from renal cancer patients and paraneoplastic controls were collected from 30 individuals who underwent surgical procedures at the Department of Urology, Shengjing Hospital of Chinese Medical University. All cases were histopathologically confirmed as clear cell renal cell carcinoma (ccRCC).

### Antibodies and reagents
The following antibodies were used for Western blot analysis: SPI1 (55100-1-AP, Proteintech), MCL1 (16225-1-AP, Proteintech, 1:1000), H3K27me3 (9733S, CST, 1:1000), Bcl-2 (68103-1-Ig, Proteintech, 1:1000), ACSL4 (22401-1-AP, Proteintech, 1:1000), EZH2 (21800-1-AP, Proteintech, 1:1000), β-ACTIN (66009-1-Ig, Proteintech, 1:2000) and GAPDH (60004-1-Ig, Proteintech, 1:2000). The anti-SPI1 antibody (55100-1-AP, Proteintech 1:500) and anti-ACSL4 antibody (22401-1-AP, Proteintech, 1:500) for IHC were purchased from Proteintech. The Co-IP experimental antibodies were anti-SPI1 antibody (2266, CST, 1:100) and EZH2 antibody (39076, mAb, 1:100). The anti-H3K27me3 antibody (9733S), the anti-SPI1 antibody (2266) and EZH2 antibody (5246S) for ChIP-PCR were purchased from CST. Erastin (HY-15763), ferrostatin-1 (HY-100579), UNC1999 (HY-15646) and Entinostat (HY-12163) was purchased from MCE.

### Small interfering RNA (siRNA) transfection
The siRNAs for SPI1, EZH2 and ACSL4 were purchased from GenePharma (Suzhou, China), and their sequences are provided in Supplementary Data 1. Briefly, $1.1–1.4 \times 10^6$ cells were cultivated into 6 cm dishes, followed by the addition of transfection reagents (PolyPlus, Shanghai, China) and siRNA (Suzhou, China) according to the manufacturer's instructions, which were collected 47–72 hours later for subsequent assays.

### Lentiviral vector construction and transfection
The lentivirus vector pHBLV-U6-MCS-CMV-ZsGreen-PGK-PURO was utilized to construct shRNAs. 786-O and ACHN cell lines were infected with lentivirus containing either shRNA-SPI1 or control shRNA (shCtrl). The specific shRNA sequences can be found in Supplementary Data 1. Following transfection of 786-O and ACHN cells with lentivirus, puromycin was used for selection, and transfection efficiency was confirmed via western blot analysis prior to subsequent experiments.

### RNA extraction and quantitative real-time PCR (qRT–PCR)
Total RNA from cells and tissues was extracted using TRIzol reagent (TianGen, China). We utilised a reverse transcription kit (TianGen, Beijing, China) to reverse transcribe the total RNA according to the instructions and then perform qRT-PCR using a PCR kit (SYBR Green). The primer sequences are provided in Supplementary Data 1. The results were analyzed by the $2^{-\Delta\Delta Ct}$ method to quantify fold changes.

### Western blot analysis
Briefly, proteins were tested for concentration after collection using RIPA buffer (Beyotime, Shanghai, China) containing protease inhibitors. Next, proteins were separated by sodium dodecyl sulfate–polyacrylamide gel electrophoresis (SDS–PAGE) and were transferred to polyvinylidene fluoride (PVDF) membranes. The membrane was blocked with 5% skimmed milk for 2 hours at room temperature and then incubated with the corresponding primary antibody at 4 °C overnight. The next day incubate with the corresponding secondary antibody for 2 hours and then expose for development.

### Immunohistochemistry (IHC)
Briefly, paraffin-embedded 4 μm-thick ccRCC tissue sections were deparaffinised and hydrated with ethanol. Subsequently, the sections were blocked with 5% BSA for 2 hours at room temperature, the sections were incubated overnight with the corresponding antibodies at 4 °C. Peroxidase conjugates were incubated in sections for 30 min at room temperature,

followed by addition of DAB substrate (Absin, shanghai, China) for visualization.

## CCK-8 cell proliferation assay

Cell activity was assessed using the CCK-8 assay kit from MCE in China. Renal cancer cells were either treated with drugs or transfected and then incubated overnight in 96-well plates at a density of 1500 cells per well. Following cell attachment, 10 μL of CCK8 solution was added to each well (containing 90 μL of culture medium) at different time points (0, 24, 48, and 72 hours). The optical density (OD) absorbance values were measured at 450 nm using a microplate reader.

## Lipid ROS assay

Lipid ROS were detected in renal cancer cells and nude mouse tumors using C11-BODIPY (C10445, Invitrogen) according to the manufacturer's directions. Briefly, treated renal cancer cells or nude mouse tumors were added to the dye under light-avoidance conditions for 30 min. After three subsequent cold PBS washes, C11-BODIPY green/red fluorescence (510 nm and 590 nm) was detected by flow cytometry and fluorescence microscopy.

## Iron assay

The concentration of $Fe^{2+}$ in tumor cells and tissues was quantified using an Iron Assay Kit (ab83366, Abcam) following the provided instructions. Tumor cells or well-milled tissues were thoroughly mixed in 100 μl of iron assay buffer and centrifuged. A 1% standard curve was prepared with 10 μl standard iron and 990 μl dd water. Subsequently, 5 μl of iron-reducing agent was added to each well of a 96-well plate and incubated at room temperature for 30 minutes. Then, 100 μl of iron probe was added to each well and incubated for 1 hour at room temperature in the absence of light. Absorbance was measured at 593 nm (A593) for visualization.

## MDA assay

The concentration of malondialdehyde (MDA) in tumor cells and tissues was quantified using an MDA Assay Kit (S0131S, Beyotime) following the provided protocol. Tumor cells ($1.5 \times 10^6$) were treated, mixed with 0.15 ml of lysate, and centrifuged at $10,000 \times g$ for ten minutes to collect the supernatant. The MDA assay working solution was prepared by combining TBA 1500 μl dilution solution, TBA 500 μl storage solution, and 30 μl antioxidant. Standard solutions were created by diluting to 1, 2, 5, 10, 20, and 50 μM for generating standard curves. Experimental, standard, and PBS solutions (100 μl each) were mixed with 200 μl of MDA assay solution, heated for 15 minutes, and then centrifuged at $1000 \times g$ for 10 minutes at room temperature. The absorbance of the resulting supernatant was measured at 532 nm using a 96-well plate.

## Chromatin immunoprecipitation assay (ChIP)

The chromatin immunoprecipitation assay was performed using a ChIP Kit from Cell Signal Technology (Cat. No. #9005, Danvers, MA, USA) according to the manufacturer's instructions. Briefly, the treated tumor cells were fixed using paraformaldehyde and subsequently broken using an ultrasonic crusher to break up the tumor cells. SPI1, EZH2 and H3K27me3 antibodies were added to the corresponding samples so that the antibody-target protein-DNA complexes were well mixed and incubated overnight. The following day, Protein A Agarose was added to the sample to precipitate the antibody-target protein-DNA complex. Subsequently, the complexes were eluted after non-specific binding was removed. Finally, DNA was purified by extraction with LiCl, phenol/chloroform, and ethanol, and the specific ACSL4 promoter region was amplified with a DNA template. The detailed sequences of the primers are listed in Supplementary Data 2.

## Coimmunoprecipitation (Co-IP)

A Co-IP assay was performed using a Co-IP Kit from Thermo Fisher (USA) following the manufacturer's instructions. Renal cancer cells were treated and then centrifuged at 1400 rpm for 3 minutes at 4 °C, after which the upper layer of culture medium was discarded. The samples were mixed with

non-denaturing lysate NETN and incubated on ice for 15 minutes. Following this, the mixture was centrifuged at $12,000 \times g$ for 5 minutes at 4 °C. Approximately 20 μL of cell lysate supernatant was mixed with 2× loading buffer and boiled for 5 minutes as the input group. S beads agarose beads were placed in new EP tubes, and an antibody against protein A was added along with the cell lysate supernatant, followed by incubation on a shaking bed at 4 °C for 2 hours. The beads were then washed three times and boiled for 5 minutes with 2× loading buffer for the IP or Co-IP group. Finally, various samples were collected for western blot analysis.

## Plasmid construction and luciferase assay

The SPI1 overexpressing plasmid pcDNA3.3-SPI1 and the plasmid pGL4.18WT+binding site (BS1 and BS2) containing the promoter region of ACSL4 were constructed. 293 T Cells cultured in 6-well plates were transfected with overexpression plasmids or empty vectors. Site-directed mutagenesis of ACSL4 was performed using the QuikChange XL system (Stratagene, La Jolla, CA, USA). The primer sequences used for the generation and confirmation of the mutant are listed in Supplementary Data 4. Luciferase activity was measured using a dual luciferase reporter system according to the manufacturer's instructions (E1910, Promega, USA).

## Immunofluorescence (IF)

The treated renal cancer cells were washed three times with PBS and fixed with 4% paraformaldehyde for 20 minutes. Subsequently, the samples were washed three more times with PBS and infiltrated in PBS with 0.1% Triton X-100 for 20 minutes. After washing the samples three times again with PBS, the cells were blocked with 10% BSA in PBS for 30 minutes. The corresponding SPI1 and EZH2 primary antibodies were incubated overnight at 4 °C. The next day, after washing the samples three times with PBST, they were incubated with the corresponding secondary antibodies for 1 hour, then sealed and photographed.

## Xenograft tumor growth

According to the experimental needs, we randomly divided 4-week-old male nude mice into four groups and raised them for a week to adapt to the environment and eliminate interference. Subsequently, we injected $2 \times 10^7$ 786-O cell line with shSPI1 or shCtrl into the right back of all nude mice. Six days later, the tumor on the back of the nude mouse was clearly visible. We measured the size of the tumor and recorded it every three days. UNC1999 is administered as a single intraperitoneal injection of 50 mg/kg once every three days. Erastin was intratumorally injected into the mice at a dose of 5 mg/kg once every three days. After 24 days, we killed the nude mice by cervical dissection, collected the subcutaneous tumors, and measured their size, volumes and weight.

The animal protocol complied with the regulations of the Animal Ethics Committee of China Medical University.

## Database application

The differentially expressed genes data between renal cancer and normal renal tissues of TCGA-KIRC was from GEPIA (| Log2FC | ≥ 1, q < 0.01) (http://gepia.cancer-pku.cn/).Kaplan-Meier survival curves generated from GEPIA database for SPI1, EZH2 and ACSL4. TCGA-KIRC data set, including normalized RNA-sequencing data and clinical data, was directly downloaded from the TCGA database, which are provided in Supplementary Data 1. Transcription factor data set were derived from the ENCODE database (http://compbio.mit.edu/encode-motifs/). The possible binding sites of transcription factor SPI1 and EZH2 to the ACSL4 promoter were predicted by the HumanTFDB database, which was listed in Supplementary Data 1.

## Statistical analysis

GraphPad Prism (version: 7.0) was used for statistical data analysis. Experimental data are expressed as the mean ± standard deviation (mean ± SD) deviation. Data were analyzed by *t* test (two groups'

comparison) and Kruskal–Wallis test (three or more groups' comparison). *: P < 0.05, **: P < 0.01, ***: P < 0.001.

## Ethics approval and consent to participate
This study was conducted at Shengjing Hospital of China Medical University in Shenyang, Liaoning Province, China, with the authorization of the Institutional Review Board of China Medical University, and the subjects gave informed consent for this study.

## Reporting summary
Further information on research design is available in the Nature Portfolio Reporting Summary linked to this article.

## Data availability
All data analyzed during this study are included in Supplementary Data. Supporting data related to this work are available upon request.

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

## Acknowledgements
We would like to thank the TCGA database and UCSC database, as well as the Department of Urology, the Shengjing Hospital of China Medical University. Our graphical abstract and some illustrations are from FigDraw, which is authorized under copyright numbers YRPIW3a859 and USRPP574ea.

## Competing interests
The authors declare no competing interests.
