## [Transparent Peer Review file · Communications Biology]

Therapeutic targeting SPI1 in combination with erastin promotes ferroptosis in ccRCC

Corresponding Author: Dr Xiang Fei

Version 0:

Reviewer comments:

Reviewer #1

(Remarks to the Author)

Therapeutic targeting SPI1 in combination with erastin promotes ferroptosis in ccRCC

In this study, Xue et al. investigate the role of the transcription factor SPI1 (PU.1) in mediating resistance to ferroptosis in clear cell renal cell carcinoma (ccRCC). The authors demonstrate that SPI1 is overexpressed in ccRCC and correlates with poor clinical outcomes. Mechanistically, SPI1 represses the lipid peroxidation gene ACSL4 via recruitment of the epigenetic regulator EZH2 and H3K27me3 modification, thereby suppressing ferroptotic cell death. Functional experiments show that SPI1 knockdown or pharmacologic inhibition of EZH2 with UNC1999—particularly when combined with the ferroptosis inducer erastin—enhances ferroptosis and reduces tumor growth in vitro and in vivo. These findings identify SPI1 as a novel ferroptosis suppressor and propose a combinatorial epigenetic-ferroptosis therapeutic strategy for ccRCC.

Overall impression of the work

This is a timely and well-designed study that addresses a significant unmet need in ccRCC, a cancer subtype characterized by resistance to conventional therapies and poor prognosis. The authors establish SPI1, a transcription factor primarily studied in hematologic malignancies, as a key regulator of ferroptosis resistance in ccRCC. They further reveal a mechanistic axis involving SPI1-mediated recruitment of EZH2 to silence ACSL4 expression epigenetically, thus linking transcriptional control to ferroptotic suppression. The study is supported by a comprehensive body of in vitro and in vivo data, including patient-derived models and xenografts. The experimental approach is rigorous, and the data convincingly support the proposed mechanistic framework. Notably, the therapeutic combination of EZH2 inhibition and ferroptosis induction is well substantiated and holds translational potential. Nevertheless, several mechanistic and methodological aspects require clarification. In particular, the specificity of the SPI1–EZH2 interaction, potential off-target effects of UNC1999 and RNAi strategies, and the clinical relevance of SPI1/ACSL4 as biomarkers warrant deeper discussion or additional validation. Addressing these points will significantly strengthen the manuscript's impact and translational relevance.

Specific comments with recommendations

1.) Mechanistic clarity of SPI1-EZH2 interaction

The manuscript presents co-immunoprecipitation data suggesting a physical interaction between SPI1 and EZH2, which supports the central hypothesis that SPI1 recruits EZH2 to repress ACSL4 transcription. However, the nature of this interaction—whether it is direct or mediated through other cofactors—remains unresolved. Providing more mechanistic insight into how SPI1 engages EZH2 would substantially strengthen the claim of a functional regulatory complex. Specifically, domain-mapping studies using truncation mutants of SPI1 and/or EZH2, along with reciprocal co-IP experiments, could help delineate the interaction interface and confirm the directness of the association.

2.) Direct vs. indirect transcriptional repression of ACSL4

The ChIP-PCR results convincingly demonstrate that both SPI1 and EZH2 are enriched at the ACSL4 promoter region, suggesting that SPI1 may participate in the epigenetic silencing of ACSL4 through recruitment of EZH2. However, the current data do not definitively establish whether SPI1 directly recruits EZH2 to the chromatin or whether the co-localization is mediated through an intermediary protein complex. This distinction is critical for fully understanding the transcriptional

repression mechanism. A ChIP-reChIP (sequential chromatin immunoprecipitation) experiment would provide strong evidence for co-occupancy of SPI1 and EZH2 at the same genomic loci and help substantiate the proposed recruitment model. If additional experimentation is not feasible, the authors should at least address this ambiguity explicitly in the Discussion section to provide a balanced interpretation of the findings and avoid overstating the mechanistic link.

3.) Specificity of UNC1999 effects

The manuscript uses UNC1999 to pharmacologically inhibit EZH2 and demonstrates that this sensitizes ccRCC cells to ferroptosis, presumably by reversing the epigenetic repression of ACSL4. However, UNC1999 is a dual inhibitor targeting both EZH2 and EZH1, and the manuscript assumes EZH2 is the sole relevant mediator without addressing the potential role of EZH1. To strengthen the mechanistic interpretation, it is important to clarify whether EZH1 is expressed in the ccRCC models used and, if so, whether it might also contribute to ACSL4 repression or compensate for EZH2 inhibition. A brief discussion of EZH1's expression profile, or ideally, supporting expression data, would improve the specificity of the conclusions and rule out potential confounding effects associated with dual inhibition.

4.) Functional rescue experiments with ACSL4 overexpression

The study presents compelling evidence that knockdown of ACSL4 can reverse the ferroptosis-sensitizing effects observed upon SPI1 inhibition, supporting the role of ACSL4 as a critical downstream effector in this regulatory axis. However, the reverse experiment—whether enforced overexpression of ACSL4 alone is sufficient to restore ferroptosis sensitivity in the presence of high SPI1 levels—has not been addressed. Such a gain-of-function approach would provide strong, complementary evidence for the functional relevance of ACSL4 repression in mediating ferroptosis resistance. Including or discussing these data would enhance the causal link between SPI1 activity, ACSL4 expression, and ferroptotic susceptibility, thereby strengthening the mechanistic model proposed in the manuscript.

5.) Clinical significance and patient stratification

The observed correlation between high SPI1 expression and poor patient prognosis in ccRCC is a key strength of the study, underscoring the potential clinical relevance of targeting the SPI1–EZH2–ACSL4 axis. However, the translational implications of this finding remain underexplored. Specifically, it would be valuable to discuss whether SPI1 or ACSL4 expression levels could serve as predictive biomarkers to stratify patients who might benefit from therapies combining ferroptosis inducers with EZH2 inhibitors. Incorporating such a perspective would enhance the clinical impact of the work and align the preclinical findings with precision oncology approaches. A brief discussion of existing expression data from patient cohorts or prospective strategies for biomarker-guided patient selection would strengthen the translational relevance of the study.

6.) Potential off-target effects

Given the central role of SPI1 in the study's mechanistic and therapeutic model, the validity of the knockdown approach is critical. While the current data support a link between SPI1 inhibition and enhanced ferroptosis sensitivity, reliance on a single siRNA or shRNA raises concerns about potential off-target effects, particularly in light of SPI1's broad regulatory functions. To strengthen the reliability of the findings, it would be advisable to validate the key phenotypes using at least one additional, independent siRNA/shRNA sequence. Alternatively, the use of orthogonal gene-silencing approaches, such as CRISPR interference, would provide robust confirmation of target specificity and help exclude confounding effects. This additional validation would enhance the rigor and reproducibility of the conclusions drawn.

7.) Safety concerns of combined therapy

The combination of UNC1999 and erastin shows promising anti-tumor effects in ccRCC models; however, as acknowledged by the authors, the potential for off-target or systemic toxicity is an important concern, especially given the non-selective nature of ferroptosis induction and EZH2 inhibition. To enhance the translational depth of the study, it would be valuable to include a more detailed discussion of strategies to mitigate these risks. For instance, the use of targeted drug delivery systems—such as tumor-specific nanoparticles, liposomal encapsulation, or antibody-drug conjugates—could limit off-target exposure and enhance therapeutic index. Additionally, localized drug administration strategies (e.g., intratumoral injection) or prodrug formulations could further reduce systemic toxicity. Addressing these options would strengthen the clinical feasibility of the proposed therapeutic strategy and anticipate key challenges in its translation to human studies.

8.) Statistical transparency

Although the manuscript reports p-values throughout the figures, key statistical details are inconsistently presented. In several instances, the definitions of error bars (e.g., standard error of the mean vs. standard deviation), the number of biological replicates (n), and the type of statistical tests applied are not clearly indicated. For full transparency and to ensure reproducibility, it is essential to provide this information uniformly across all figure legends. Each experimental condition should clearly state the sample size (n), the nature of the error bars, and the exact statistical test used, including whether tests were one-sided or two-sided.

Minor comments

- 1.) Please correct the typos in several places: e.g., "is not well clear" → "is not well understood".
- 2.) Define all abbreviations upon first use, e.g., "MDA" and "Fer-1".
- 3.) Ensure consistent figure referencing in the main text (e.g., "Fig.3K" not "Fig. 3K").

Reviewer #2

(Remarks to the Author)

Dr. Fei and colleagues have authored an article that highlights the significance of targeting SPI1 in combination with erastin, shedding light on its underlying mechanisms. Their study demonstrates that this combination effectively promotes ferroptosis in renal carcinoma. The manuscript is well-written, hypothesis-driven, and systematically follows through on its proposed framework. Importantly, it holds immense translational value with potential therapeutic implications.

However, a few key points need to be addressed before it is ready for acceptance:

1. Recent studies have shown that targeting glutamine dependency increases vulnerability to GPX4-dependent ferroptosis (PMID: 39879992, PMID: 31911550). The authors should include a brief discussion of this aspect in the Introduction to provide better context for their work.
2. Additionally, commenting on the potential use of the GPX4 inhibitor RSL3 in future studies would be a valuable addition. This could enhance the future perspectives of the study, particularly in relation to ACSL4 and SPI1.
3. It is well established that SPI1 is a substrate of Keap1 and plays a role in regulating NRF2 activity (PMID: 33895141, PMID: 34681893). This aligns with the mechanism discussed in point 1. The authors should incorporate a brief discussion of this in the Discussion section to strengthen their argument.

Version 1:

Reviewer comments:

Reviewer #2

(Remarks to the Author)

Accepted

Response to the Editor

Communications biology

COMMSBIO-24-8506A

Therapeutic targeting SPI1 in combination with erastin promotes ferroptosis in ccRCC

Dear Editors,

Thank you for providing us this opportunity to further revise our manuscript. We also appreciate the thoughtful and constructive comments from the editors and reviewers and have taken great care to incorporate their suggestions into the revised manuscript. We have submitted our revised manuscript entitled “Therapeutic targeting SPI1 in combination with erastin promotes ferroptosis in ccRCC” to editorial office by online submission system. In accordance with the manuscript policy requirements, we have enhanced all charts depicting single values (such as averages) with error bars, and all icons now display data distribution. We performed molecular weight marker labeling for all the blot/gel images. Supplementary data files (in Excel format) providing the numerical source data for charts and graphs are included in the supplementary materials section. We have also included uncropped and unedited blot/gel images as supplementary figures in the PDF file of supplementary information. We also submitted a rebuttal letter, a marked-up version of the manuscript, a clean version of the manuscript and a Reporting Summary. According to the comments from the reviewers, we have carefully amended the manuscript and answered the questions point by point. Changes in the revised manuscript were marked in yellow.

According to the reviewers' comments, we have supplemented extensive experiments for validation. Jixin Li has made significant contributions during the revision process. We would like to respectfully inquire whether it would be possible to include him as a co-first author. We have consulted and obtained consent from all authors. Of course, we highly respect your suggestions and the journal's requirements. If you approve, we will make this modification during the next revision. If this arrangement is deemed inappropriate, we will not implement any changes.

Finally, thank you for your contributions to improving the quality of the manuscript.
Wishing you a pleasant work experience.

Yours sincerely

Xiang Fei

Dear reviewers,

We sincerely appreciate your comments and suggestions, which have helped improve the quality of our manuscript. Based on your recommendations and requirements, we have made corresponding revisions to the manuscript. We hope our work can be further enhanced. For easy reference, the questions are marked in blue, while the responses are presented in black. Additionally, any modifications made to the original text have been highlighted in yellow with corresponding line numbers indicated. Furthermore, we would like to show the details as follows:

Response to Reviewer 1

Major points:

Q1.) Mechanistic clarity of SPI1-EZH2 interaction

The manuscript presents co-immunoprecipitation data suggesting a physical interaction between SPI1 and EZH2, which supports the central hypothesis that SPI1 recruits EZH2 to repress ACSL4 transcription. However, the nature of this interaction—whether it is direct or mediated through other cofactors—remains unresolved. Providing more mechanistic insight into how SPI1 engages EZH2 would substantially strengthen the claim of a functional regulatory complex. Specifically, domain-mapping studies using truncation mutants of SPI1 and/or EZH2, along with reciprocal co-IP experiments, could help delineate the interaction interface and confirm the directness of the association.

A1- Thank you for your comments. Based on your suggestions, we first predicted the molecular interaction domains of SPI1 and EZH2. The sequences of EZH2 and SPI1 were submitted to the alphafold3 server.

Information

Type	Copies	Sequence
Protein	1	<pre> 10 20 30 40 50 60 MGQTGKKSEK GPVCWRKRVK SEYMLRLQLK RFRRRADEVKS MFSSNRQKIL ERTEILNQEW 70 80 90 100 110 120 KORRIQPVHI LTSVSSLRGT RECSVTSOLD FPTQVIPLKT LNAVASVPI YSWSPLOQNF 130 140 150 160 170 180 MVEDETVLHN IPYMGDEVLD QDGTFIEELI KNYDGKVHGD RECGFINDEI FVELVNALGO 190 200 210 220 230 240 YNDDDDDDDG DDPEEREKQ KDLEDHRDDK ESRPPRKFP DKIFEAISSM FPDKGTAEEEL 250 260 270 280 290 300 KEKYKELTEQ QLPGALPPEC TPNIDGPNAK SVQREOSLHS FHTLFCRRCF KYDCFLHPPH 310 320 330 340 350 360 ATPNTYKRKN TETALDNKPC GPQCYQHLEG AKEFAAALTA ERIKTPPKRP GGRRRGRLPN 370 380 390 400 410 420 NSSRPSPTI NVLESKDTDS DREAGTETGG ENNDKEEEEK KDETSSSSEA NSRCQTIK M 430 440 450 460 470 480 KPNIIPPENV EWSGAEASMF RVLIGTYYDN FCAIARLIGT KTCRQVYEF VKESSIIAPA 490 500 510 520 530 540 PAEDVDTPPR KKKRKHRLWA AHCRIQLKK DGSSNHVYNY QPCDHPRPC DSSPCVIAQ 550 560 570 580 590 600 NFCCKFCQCS SECQNRFPGC RCKAQCNFKQ CPCYLAVREC YPDLCLCTGA ADHWDSKNVS 610 620 630 640 650 660 CKNCISIQRS KKHLLAPSD VAGWGIFIKD PVQKNEFISE YCGEIIISDE ADRRGKVYDK 670 680 690 700 710 720 YMCSFLFNLN NDFVVDATRK GNKIRFANHS VNPNCYAKVM MVNGDHRIGI FAKRAIQTGE 730 740 746 ELFFDYRYSQ ADALKYVGIE REMEIP </pre>
Protein	1	<pre> 10 20 30 40 50 60 MLQACKMEGF PLVPPPSDL VPYDLDLYQR QTHEYYPYLS SDGESHSDDHY WDFHPHHVHS 70 80 90 100 110 120 EFESFAENNF TELQSVOPPQ LQQLYRHMEL EQMHVLDTPM VPPHPSLGHO VSYLPRMCLO 130 140 150 160 170 180 YPSLSPAQPS SDEEEGERQS PPLEVSDGEA DGLEPGPGLL PGETGSKKKI RLYQFLDLLL 190 200 210 220 230 240 RSGDMKDSIW WVDKDKGTFQ FSSKHKEALA HRWGIQKGNR KKMTYQKMAR ALRNYGKTGE 250 260 270 VKKVKKCLTY QFSGEVLGRG GLAERRHPPH </pre>

Seed: 681535128

Describe the amino acid sequences of EZH2 protein and SPI1 protein

Through the computation of the AlphaFold3 server, we obtained the predicted model of the protein complex. The model provided a predicted Template Modeling score (pTM) = 0.31, an interface Template Modeling score (ipTM) = 0.52, and a combined score (pTM+ipTM) = 0.84. This score significantly exceeds our set interaction threshold of 0.6, indicating a high potential for interaction between the two proteins and a high likelihood of interaction.

EZH2-SPI1

Calculate the interaction score between EZH2 and SPI1 proteins using the AlphaFold3 server

To further validate the accuracy of the prediction results, we used PyMol software to continue analyzing the result files and generated detailed images of the binding sites of two proteins, where the EZH2 protein is shown in green and the SPI1 protein in cyan. The yellow dashed lines in the images indicate the amino acids involved in hydrogen bond interactions.

The binding sites of EZH2 protein (green) and SPI1 protein (cyan) were analyzed using PyMol software, with yellow dashed lines indicating the amino acids involved in hydrogen bond interactions

Using the prosite tool to predict the domains of EZH2 and SPI1 respectively, it was found that the EZH2 protein contains two domains: the CXC domain (amino acids 503-605) and the SET domain (amino acids 612-727). The SPI1 protein has one domain, the EST-DOM (amino acids 170-253). Based on the aforementioned binding sites, it can be inferred that the interacting domains of the two proteins are primarily the SET domain of EZH2 and the EST-DOM domain of SPI1.

To further validate the interaction between the SET domain of EZH2 and the EST-DOM domain of SPI1, we tagged the wild-type and truncated mutant (deletion of 612-727aa) of EZH2 protein with a Flag tag, and the wild-type and truncated mutant (deletion of 170-253aa) of SPI1 protein with a His tag. We then co-transfected the two types of EZH2 plasmids with either the wild-type or mutant SPI1 into the 293T cell line for co-IP assay (A-C). The results showed that the interaction was lost after truncation. These findings indicate that EZH2 and SPI1 can directly interact through their SET and EST-DOM domains.

(A) The corresponding amino acid sequences of EZH2 and SPI1 were truncated by mutation, and the mutants were labeled with Flag and His tags, respectively. (B, C) Co-IP assay was performed to analyze the interaction between EZH2 wild-type or mutant and SPI1 wild-type or mutant

Q2.) Direct vs. indirect transcriptional repression of ACSL4

The ChIP-PCR results convincingly demonstrate that both SPI1 and EZH2 are enriched at the ACSL4 promoter region, suggesting that SPI1 may participate in the epigenetic silencing of ACSL4 through recruitment of EZH2. However, the current data do not definitively establish whether SPI1 directly recruits EZH2 to the chromatin or whether the co-localization is mediated through an intermediary protein complex. This distinction is critical for fully understanding the transcriptional repression mechanism. A ChIP-reChIP (sequential chromatin immunoprecipitation) experiment would provide strong evidence for co-

occupancy of SPI1 and EZH2 at the same genomic loci and help substantiate the proposed recruitment model. If additional experimentation is not feasible, the authors should at least address this ambiguity explicitly in the Discussion section to provide a balanced interpretation of the findings and avoid overstating the mechanistic link.

A2- Thank you for your reminder and valuable advice. As you pointed out, ChIP-PCR only indicates the enrichment of SPI1 and EZH2 in the promoter region of ACSL4, but it does not demonstrate whether SPI1 recruits EZH2 to the promoter region directly or through intermediary proteins. I have included relevant explanations in the discussion section of the original text (Line 461-466).

Q3.) Specificity of UNC1999 effects

The manuscript uses UNC1999 to pharmacologically inhibit EZH2 and demonstrates that this sensitizes ccRCC cells to ferroptosis, presumably by reversing the epigenetic repression of ACSL4. However, UNC1999 is a dual inhibitor targeting both EZH2 and EZH1, and the manuscript assumes EZH2 is the sole relevant mediator without addressing the potential role of EZH1. To strengthen the mechanistic interpretation, it is important to clarify whether EZH1 is expressed in the ccRCC models used and, if so, whether it might also contribute to ACSL4 repression or compensate for EZH2 inhibition. A brief discussion of EZH1's expression profile, or ideally, supporting expression data, would improve the specificity of the conclusions and rule out potential confounding effects associated with dual inhibition.

A3- We are very grateful for your comments and corrections. Compared with EZH2, EZH1 shows lower expression in ccRCC. We performed qRT-PCR analysis on 30 pairs of ccRCC and adjacent tissue samples to assess the expression of EZH1 and EZH2. The results indicated that EZH2 is more abundantly expressed in ccRCC.

qRT-PCR analysis of EZH1 and EZH2 mRNA expression levels in 30 pairs of renal carcinoma and adjacent tissues (n=30). Data are presented as the mean \pm SD and analyzed by t-test or Kruskal-Wallis test. *: P<0.05, **: P<0.01, ***: P<0.001

To investigate whether EZH1 has a regulatory effect on ACSL4, we knocked down EZH1 in the 786-O and ACHN cell lines. There were no significant changes in either ACSL4 mRNA or protein levels, suggesting that EZH1 may not have a significant regulatory role on ACSL4 in ccRCC.

In the 786-O and ACHN cell lines, knockdown of EZH2 combined with overexpression of EZH1 demonstrated that EZH1 cannot substitute for EZH2 in regulating ACSL4.

Western Blot analysis of ACSL4 protein expression in the corresponding groups of 786-O and ACHN cell lines

The above experiments demonstrate that EZH1 is lowly expressed in ccRCC and exhibits minimal regulatory effect on ACSL4, thus essentially excluding potential

confounding effects related to dual inhibition. EZH2 and EZH1, as the core catalytic subunits of the PRC2 complex, although homologous and both capable of catalyzing H3K27me3, exhibit independent mechanisms in different biological processes. For instance, in breast cancer metastasis, the methylation modification of EZH2 (such as R342 site methylation mediated by PRMT1) enhances its protein stability, promoting tumor cell invasion and metastasis, while EZH1 is not involved in this process [1]. In neuropathic pain, EZH2 drives pain occurrence by inhibiting the expression of TIMP1 in spinal GABAergic interneurons, activating the MMP-9-TLR2/4-NLRP3 signaling pathway, a mechanism entirely dependent on EZH2 and unrelated to EZH1 [2]. EZH2 exacerbates autoimmune inflammatory responses by promoting the activation of the NF- κ B signaling pathway through the suppression of the anti-inflammatory gene Socs3 in macrophages, a process in which the function of EZH2 is not replaced or compensated by EZH1 [3]. The above experiments and references indicate that EZH2 regulates ACSL4 expression through the H3K27me3 pathway independently of EZH1 in ccRCC.

Q4.) Functional rescue experiments with ACSL4 overexpression

The study presents compelling evidence that knockdown of ACSL4 can reverse the ferroptosis-sensitizing effects observed upon SPI1 inhibition, supporting the role of ACSL4 as a critical downstream effector in this regulatory axis. However, the reverse experiment—whether enforced overexpression of ACSL4 alone is sufficient to restore ferroptosis sensitivity in the presence of high SPI1 levels—has not been addressed. Such a gain-of-function approach would provide strong, complementary evidence for the functional relevance of ACSL4 repression in mediating ferroptosis resistance. Including or discussing these data would enhance the causal link between SPI1 activity, ACSL4 expression, and ferroptotic susceptibility, thereby strengthening the mechanistic model proposed in the manuscript.

A4- Thanks to your findings and reminders. We constructed SPI1 overexpression plasmids and transfected them into the Caki-1 and A498 cell lines, finding that SPI1 overexpression inhibits ACSL4 expression (A). The CCK-8 assay demonstrated that SPI1 overexpression increases tumor cell activity (B). FCM verified that SPI1 overexpression reduces ROS levels in the tumor cell lines (C). Additionally, SPI1 overexpression can also decrease the accumulation of Fe²⁺ and MDA levels (D, E).

(A) Western Blot analysis of ACSL4 protein expression in the corresponding groups of Caki-1 and A498. (B) CCK-8 assays were used to analyse the effect on cell viability in Caki-1 and A498. (C) Flow cytometry was used to detect lipid peroxidation level in Caki-1 and A498 (n=3). (D, E) MDA and Fe²⁺ levels were measured separately in 786-O and ACHN (n=4). Data are presented as the mean \pm SD and analyzed by t-test or Kruskal-Wallis test. *: P<0.05, **: P<0.01, ***: P<0.001

To further verify that SPI1 regulates the ACSL4 target through EZH2 to influence the ferroptosis phenotype, knocking down EZH2 in SPI1-overexpressing cells abolished the inhibitory effect of SPI1 overexpression on ACSL4 (A). The CCK-8 assay demonstrated that overexpression of ACSL4 could reverse the ferroptosis inhibition caused by SPI1 overexpression (B). FCM confirmed that overexpression of ACSL4 could rescue the downregulation of ROS levels induced by SPI1 overexpression (C). Similarly, the downregulation of MDA and Fe²⁺ levels due to SPI1 overexpression could also be reversed by ACSL4 overexpression (D, E).

In summary, under high SPI1 levels, forced overexpression of ACSL4 can restore sensitivity to ferroptosis, indicating that the reverse validation of SPI1 synergistically regulating ACSL4 expression with EZH2 also holds true.

(A) Western Blot analysis of ACSL4 and EZH2 protein expression in the corresponding groups of Caki-1 and A498. (B) CCK-8 assays were used to analyse the effect on cell viability in Caki-1 and A498. (C) Flow cytometry was used to detect lipid peroxidation level in Caki-1 and A498 (n=3). (D, E) MDA and Fe²⁺ levels were measured separately in 786-O and ACHN (n=3). Data are presented as the mean \pm SD and analyzed by t-test or Kruskal-Wallis test. *: P<0.05, **: P<0.01, ***: P<0.001

Q5.) Clinical significance and patient stratification

The observed correlation between high SPI1 expression and poor patient prognosis in ccRCC is a key strength of the study, underscoring the potential clinical relevance of targeting the SPI1–EZH2–ACSL4 axis. However, the translational implications of this finding remain underexplored. Specifically, it would be valuable to discuss whether SPI1 or ACSL4 expression levels could serve as predictive biomarkers to stratify patients who might benefit from therapies combining ferroptosis inducers with EZH2 inhibitors. Incorporating such a perspective would enhance the clinical impact of the work and align the preclinical

findings with precision oncology approaches. A brief discussion of existing expression data from patient cohorts or prospective strategies for biomarker-guided patient selection would strengthen the translational relevance of the study.

A5- Thank you very much for your question. I fully agree with your viewpoint that the translation of clinical research into practice would be highly meaningful. We utilized renal cancer tissue samples from our research group to validate the mRNA and protein expression levels of SPI1 and ACSL4, and analyzed their close correlation with clinical prognosis. Meanwhile, our experimental results demonstrate that when SPI1 is highly expressed or ACSL4 is lowly expressed, the combination of ferroptosis inducers with EZH2 inhibitors can promote ferroptosis in ccRCC. This indicates that the expression levels of SPI1 or ACSL4 can serve as predictive biomarkers for identifying patient populations likely to benefit from combination therapy with ferroptosis inducers and EZH2 inhibitors. In the discussion sections of our manuscript, we have analyzed that the expression levels of SPI1 or ACSL4 could serve as predictive biomarkers (Line 480-489).

Q6.) Potential off-target effects

Given the central role of SPI1 in the study's mechanistic and therapeutic model, the validity of the knockdown approach is critical. While the current data support a link between SPI1 inhibition and enhanced ferroptosis sensitivity, reliance on a single siRNA or shRNA raises concerns about potential off-target effects, particularly in light of SPI1's broad regulatory functions. To strengthen the reliability of the findings, it would be advisable to validate the key phenotypes using at least one additional, independent siRNA/shRNA sequence. Alternatively, the use of orthogonal gene-silencing approaches, such as CRISPR interference, would provide robust confirmation of target specificity and help exclude confounding effects. This additional validation would enhance the rigor and reproducibility of the conclusions drawn.

A6- Thank you for your reminder and the careful detail. We consider your concerns to be warranted. Prior to the commencement of the experiment, we selected two specific short hairpin RNAs (shRNAs) targeting SPI1, and the knockdown efficiency is demonstrated in Result 3A. To prevent off-target effects, we further validated the relevant phenotypes of shSPI1-1, and the experimental results are as follows: Western

blot analysis showed that the expression of ACSL4 was upregulated following SPI1 knockdown, which is consistent with the results of shSPI1 (A). The CCK-8 assay confirmed that the combination of shSPI1-1 and erastin significantly inhibited tumor cell activity, and this inhibition could be reversed by the ferroptosis inhibitor Ferrostatin-1 (Fer-1) (B). Flow cytometry revealed that the combination of shSPI1-1 and erastin further promoted the level of lipid peroxidation in tumor cells (C). Additionally, we observed elevated levels of MDA and Fe^{2+} accumulation under the treatment of shSPI1-1 and erastin (D, E).

(A) Western Blot analysis of SPI1 protein expression in the corresponding groups of Caki-1 and A498. (B) CCK-8 assays were used to analyse the effect on cell viability in Caki-1 and A498. (C) Flow cytometry was used to detect lipid peroxidation level in Caki-1 and A498 (n=3). (D, E) MDA and Fe^{2+} levels were measured separately in 786-O and ACHN (n=4). Data are presented as the mean \pm SD and analyzed by t-test or Kruskal-Wallis test. *: $P < 0.05$, **: $P < 0.01$, ***: $P < 0.001$

Q7.) Safety concerns of combined therapy

The combination of UNC1999 and erastin shows promising anti-tumor effects in ccRCC models; however, as acknowledged by the authors, the potential for off-target or systemic toxicity is an important concern, especially given the non-

selective nature of ferroptosis induction and EZH2 inhibition. To enhance the translational depth of the study, it would be valuable to include a more detailed discussion of strategies to mitigate these risks. For instance, the use of targeted drug delivery systems—such as tumor-specific nanoparticles, liposomal encapsulation, or antibody-drug conjugates—could limit off-target exposure and enhance therapeutic index. Additionally, localized drug administration strategies (e.g., intratumoral injection) or prodrug formulations could further reduce systemic toxicity. Addressing these options would strengthen the clinical feasibility of the proposed therapeutic strategy and anticipate key challenges in its translation to human studies.

A7-Thank you very much for your valuable suggestions. As you mentioned, small molecule inhibitors have demonstrated promising potential in anti-tumor effects, however, mitigating their toxic reactions and off-target effects remains a significant challenge. With advancements in technology, numerous solutions have emerged. Tumor-specific nanoparticles, liposome encapsulation, and antibody-drug conjugates (ADCs) are prevalent targeted drug delivery systems. Nanoparticles can passively target tumors through the enhanced permeability and retention (EPR) effect or actively target them via surface modifications, such as folate and transferrin receptor ligands [4]. Nanoparticles modified with transferrin receptor (TfR) or PD-L1 antibodies can precisely target tumor cells, thereby minimizing the toxic effects of the drugs on the body [5]. Liposomes can prolong the drug's half-life and reduce nonspecific distribution, exemplified by irinotecan liposome (Onivyde) for the treatment of pancreatic cancer and PEGylated liposomal doxorubicin (Doxil), which significantly reduces cardiotoxicity [6, 7]. ADC-related drugs have been utilized in the treatment of various tumors, and the new generation of ADCs, such as Enhertu (DS-8201), enhances efficacy and reduces toxicity through cleavable linkers and a high drug-to-antibody ratio (DAR) [8].

Additionally, localized drug delivery strategies are crucial for mitigating drug toxicity. Directly injecting drugs into the tumor site reduces systemic exposure, thereby alleviating drug toxicity. For instance, intratumoral injection of PD-1 inhibitors combined with oncolytic viruses enhances anti-tumor immune responses [9], and intratumoral injection of the oncolytic virus T-VEC (Imlygic) is used for melanoma treatment [10]. Injectable hydrogels continuously release drugs, such as doxorubicin,

within the postoperative tumor cavity, thereby minimizing systemic diffusion [11]. Enzyme-activated prodrugs are also a key approach to addressing drug toxicity reactions. γ -Glutamyltranspeptidase (GGT)-activated prodrugs release active drugs in regions with high enzyme expression in tumors, thereby targeting tumor cells [12]. Some prodrugs, such as Tirapazamine derivatives, are activated in hypoxic or high reactive oxygen species (ROS) environments within tumors [13]. Therefore, we will conduct in vivo toxicity response-related studies in subsequent research to enhance the clinical feasibility of the proposed therapeutic strategy.

In the discussion section of the manuscript, we have made the corresponding changes and highlighted them (Line 521-539).

Q8.) Statistical transparency

Although the manuscript reports p-values throughout the figures, key statistical details are inconsistently presented. In several instances, the definitions of error bars (e.g., standard error of the mean vs. standard deviation), the number of biological replicates (n), and the type of statistical tests applied are not clearly indicated. For full transparency and to ensure reproducibility, it is essential to provide this information uniformly across all figure legends. Each experimental condition should clearly state the sample size (n), the nature of the error bars, and the exact statistical test used, including whether tests were one-sided or two-sided.

A8- We appreciate your rigorous suggestions and consider them to be highly necessary. We have accordingly annotated the figure legends in the manuscript (highlighted in yellow), including specifying the sample size (n), the nature of the error bars, and the exact statistical tests used for each experimental condition, as well as whether the tests were one-sided or two-sided (Line 724-725, 733, 743-744, 752-753, 766, 773, 783-784, 794-795).

Minor comments

Q1.) Please correct the typos in several places: e.g., "is not well clear" → "is not well understood".

A1- Thank you for your corrections and reminders. We have made the corresponding revisions in the paper abstract (Line 25), and we have also corrected some spelling details, such as correcting Fe²⁺ to Fe²⁺ in the manuscript.

Q2.) Define all abbreviations upon first use, e.g., "MDA" and "Fer-1".

Q2- Thank you very much for your suggestions. We have defined all abbreviations upon their first use in the manuscript as requested (Line 182, 305, 410).

Q3.) Ensure consistent figure referencing in the main text (e.g., "Fig.3K" not "Fig. 3K").

Q3- Thank you for your meticulous observation. We have corrected the references to the figures in the main text to ensure consistency.

References:

[1] Li Z, Wang D, Lu J, et al. Methylation of EZH2 by PRMT1 regulates its stability and promotes breast cancer metastasis. *Cell Death Differ.* 2020;27(12):3226-3242. doi:10.1038/s41418-020-00615-9

[2] Wan L, Guo H, Hu F, et al. EZH2-mediated suppression of TIMP1 in spinal GABAergic interneurons drives microglial activation via MMP-9-TLR2/4-NLRP3 signaling in neuropathic pain. *Brain Behav Immun.* Published online April 8, 2025. doi:10.1016/j.bbi.2025.04.007

[3] Zhang X, Wang Y, Yuan J, et al. Macrophage/microglial Ezh2 facilitates autoimmune inflammation through inhibition of Socs3. *J Exp Med.* 2018;215(5):1365-1382. doi:10.1084/jem.20171417

[4] Peer D, Karp JM, Hong S, Farokhzad OC, Margalit R, Langer R. Nanocarriers as an emerging platform for cancer therapy. *Nat Nanotechnol.* 2007;2(12):751-760. doi:10.1038/nnano.2007.387

[5] Singh N, Ramnarine VR, Song JH, et al. The long noncoding RNA H19 regulates tumor plasticity in neuroendocrine prostate cancer. *Nat Commun.* 2021;12(1):7349. Published 2021 Dec 21. doi:10.1038/s41467-021-26901-9

- [6] Fuhrmann G. Drug delivery as a sustainable avenue to future therapies. *J Control Release*. 2023;354:746-754. doi:10.1016/j.jconrel.2023.01.045
- [7] Barenholz Y. Doxil®--the first FDA-approved nano-drug: lessons learned. *J Control Release*. 2012;160(2):117-134. doi:10.1016/j.jconrel.2012.03.020
- [8] Modi S, Saura C, Yamashita T, et al. Trastuzumab Deruxtecan in Previously Treated HER2-Positive Breast Cancer. *N Engl J Med*. 2020;382(7):610-621. doi:10.1056/NEJMoa1914510
- [9] Ribas A, Dummer R, Puzanov I, et al. Oncolytic Virotherapy Promotes Intratumoral T Cell Infiltration and Improves Anti-PD-1 Immunotherapy [published correction appears in *Cell*. 2018 Aug 9;174(4):1031-1032. doi: 10.1016/j.cell.2018.07.035.]. *Cell*. 2017;170(6):1109-1119.e10. doi:10.1016/j.cell.2017.08.027
- [10] Champiat S, Tselikas L, Farhane S, et al. Intratumoral Immunotherapy: From Trial Design to Clinical Practice. *Clin Cancer Res*. 2021;27(3):665-679. doi:10.1158/1078-0432.CCR-20-0473
- [11] Liu R, Liang Q, Luo JQ, et al. Ferritin-Based Nanocomposite Hydrogel Promotes Tumor Penetration and Enhances Cancer Chemoimmunotherapy. *Adv Sci (Weinh)*. 2024;11(3):e2305217. doi:10.1002/advs.202305217
- [12] Li C, Penet MF, Wildes F, et al. Nanoplex delivery of siRNA and prodrug enzyme for multimodality image-guided molecular pathway targeted cancer therapy. *ACS Nano*. 2010;4(11):6707-6716. doi:10.1021/nn102187v
- [13] Ghedira D, Voissière A, Peyrode C, et al. Structure-activity relationship study of hypoxia-activated prodrugs for proteoglycan-targeted chemotherapy in chondrosarcoma. *Eur J Med Chem*. 2018;158:51-67. doi:10.1016/j.ejmech.2018.08.060

Response to Reviewer 2

Q1. Recent studies have shown that targeting glutamine dependency increases vulnerability to GPX4-dependent ferroptosis (PMID: 39879992, PMID: 31911550). The authors should include a brief discussion of this aspect in the Introduction to provide better context for their work.

A1. Thank you for the advice provided. We fully agree with your perspective that ferroptosis is essentially an imbalance in intracellular redox status. The accumulation of oxidative substances or the deficiency of reductive substances within cells can disrupt the equilibrium, leading to cell death [1]. As a reductive substance, glutathione has become a key metabolic factor in cellular ferroptosis. For the synthesis of glutathione, cysteine can be oxidized and taken up from the environment via neutral amino acid transporters or the Xc- cystine/glutamate antiporter, or synthesized through the transsulfuration pathway using methionine and glucose [2]. Current research indicates that synthesizing glutathione or enhancing the activity of the Xc- system or GPX4 can protect cells from various oxidative stresses, especially cell death induced by thiol deficiency [3-5]. Therefore, based on your suggestion, we have discussed in the introduction section of the manuscript that targeting glutamine dependency increases susceptibility to GPX4-dependent ferroptosis, and we consider the two articles you mentioned (PMID: 39879992, PMID: 31911550) to be highly novel and representative, which we have cited in the manuscript (Line 67-72).

Q2. Additionally, commenting on the potential use of the GPX4 inhibitor RSL3 in future studies would be a valuable addition. This could enhance the future perspectives of the study, particularly in relation to ACSL4 and SPI1.

A2. Thank you for your valuable suggestions. As you mentioned, the GPX4 inhibitor RSL3 holds great promise for future applications. Since ACSL4 facilitates the integration of polyunsaturated fatty acids (PUFAs) into the phospholipids of cell membranes, it enhances sensitivity to ferroptosis. RSL3, a GPX4 inhibitor, may

therefore be closely associated with ACSL4 expression levels in inducing cellular ferroptosis. Studies have indicated that tumors exhibiting high ACSL4 expression, such as certain triple-negative breast cancers and liver cancers, may demonstrate increased sensitivity to RSL3 [6]. In future research, we will employ single-cell sequencing technology to analyze the differences in sensitivity to RSL3 among various renal cancer subpopulations characterized by differing ACSL4 expression levels. We have incorporated relevant content into the discussion section of our manuscript (Line 500-507).

Q3. It is well established that SPI1 is a substrate of Keap1 and plays a role in regulating NRF2 activity (PMID: 33895141, PMID: 34681893). This aligns with the mechanism discussed in point 1. The authors should incorporate a brief discussion of this in the Discussion section to strengthen their argument.

A3. Thank you very much for your input. SPI1 and SP1 are distinct genes that, despite their similar nomenclature, exhibit significant differences in gene family classification, functional roles, and chromosomal locations. SPI1 is a member of the ETS (E26 transformation-specific) transcription factor family [7], whereas SP1 belongs to the SP/KLF (Specificity Protein/Krüppel-like Factor) transcription factor family [8]. The perspective you provided is indeed valuable. Research has demonstrated that in renal cancer, DPP9 competes with NRF2 for binding to KEAP1 in an enzyme-independent manner. This competition disrupts the KEAP1-NRF2 interaction, promotes NRF2 stability, and ultimately contributes to sorafenib resistance [9]. Currently, there is no conclusive evidence that SPI1 can function as a substrate for KEAP1. We have also discussed the research on the NRF2-KEAP1 interaction in renal cancer within the discussion section of our manuscript (Line 493-494). We greatly appreciate your suggestions and will continue to monitor the research developments regarding SPI1 and KEAP1 in the future.

[1]. Wang H, Cheng Y, Mao C, et al. Emerging mechanisms and targeted therapy of ferroptosis in cancer. *Mol Ther.* 2021;29(7):2185-2208. doi:10.1016/j.ymthe.2021.03.022

- [2]. Jiang X, Stockwell BR, Conrad M. Ferroptosis: mechanisms, biology and role in disease. *Nat Rev Mol Cell Biol.* 2021;22(4):266-282. doi:10.1038/s41580-020-00324-8
- [3]. Niu B, Liao K, Zhou Y, et al. Application of glutathione depletion in cancer therapy: Enhanced ROS-based therapy, ferroptosis, and chemotherapy. *Biomaterials.* 2021;277:121110. doi:10.1016/j.biomaterials.2021.121110
- [4]. Koppula P, Zhuang L, Gan B. Cystine transporter SLC7A11/xCT in cancer: ferroptosis, nutrient dependency, and cancer therapy. *Protein Cell.* 2021;12(8):599-620. doi:10.1007/s13238-020-00789-5
- [5]. Shen X, Chen Y, Tang Y, et al. Targeting pancreatic cancer glutamine dependency confers vulnerability to GPX4-dependent ferroptosis. *Cell Rep Med.* 2025;6(2):101928. doi:10.1016/j.xcrm.2025.101928
- [6]. Doll S, Proneth B, Tyurina YY, et al. ACSL4 dictates ferroptosis sensitivity by shaping cellular lipid composition. *Nat Chem Biol.* 2017;13(1):91-98. doi:10.1038/nchembio.2239
- [7]. Burda P, Laslo P, Stopka T. The role of PU.1 and GATA-1 transcription factors during normal and leukemogenic hematopoiesis. *Leukemia.* 2010;24(7):1249-1257. doi:10.1038/leu.2010.104
- [8]. Vizcaíno C, Mansilla S, Portugal J. Sp1 transcription factor: A long-standing target in cancer chemotherapy. *Pharmacol Ther.* 2015;152:111-124. doi:10.1016/j.pharmthera.2015.05.008
- [9]. Chang K, Chen Y, Zhang X, et al. DPP9 Stabilizes NRF2 to Suppress Ferroptosis and Induce Sorafenib Resistance in Clear Cell Renal Cell Carcinoma. *Cancer Res.* 2023;83(23):3940-3955. doi:10.1158/0008-5472.CAN-22-4001